# Genetically engineered red cells expressing single domain camelid antibodies confer long-term protection against botulinum neurotoxin

Nai-Jia Huang[1], Novalia Pishesha[1,2], Jean Mukherjee[3], Sicai Zhang[4,5,6], Rhogerry Deshycka[1,7], Valentino Sudaryo[1,2], Min Dong[4,5,6], Charles B. Shoemaker[3] & Harvey F. Lodish[1,2,7]

A short half-life in the circulation limits the application of therapeutics such as single-domain antibodies (VHHs). We utilize red blood cells to prolong the circulatory half-life of VHHs. Here we present VHHs against botulinum neurotoxin A (BoNT/A) on the surface of red blood cells by expressing chimeric proteins of VHHs with Glycophorin A or Kell. Mice whose red blood cells carry the chimeric proteins exhibit resistance to 10,000 times the lethal dose ($LD_{50}$) of BoNT/A, and transfusion of these red blood cells into naive mice affords protection for up to 28 days. We further utilize an improved CD34+ culture system to engineer human red blood cells that express these chimeric proteins. Mice transfused with these red blood cells are resistant to highly lethal doses of BoNT/A. We demonstrate that engineered red blood cells expressing VHHs can provide prolonged prophylactic protection against bacterial toxins without inducing inhibitory immune responses and illustrates the potentially broad translatability of our strategy for therapeutic applications.

[1] Whitehead Institute for Biomedical Research, Cambridge, Massachusetts 02142, USA. [2] Department of Biological Engineering, Massachusetts Institute of Technology, Cambridge, Massachusetts 02139, USA. [3] Tufts Cummings School of Veterinary Medicine, North Grafton, Massachusetts 01536, USA. [4] Department of Urology, Boston Children's Hospital, Harvard Medical School, Boston, Massachusetts 02115, USA. [5] Department of Microbiology and Immunobiology, Harvard Medical School, Boston, Massachusetts 02115, USA. [6] Department of Surgery, Harvard Medical School, Boston, Massachusetts 02115, USA. [7] Department of Biology, Massachusetts Institute of Technology, Cambridge, Massachusetts 02139, USA. Nai-Jia Huang and Novalia Pishesha contributed equally to this work. Correspondence and requests for materials should be addressed to C.B.S. (email: charles.shoemaker@tufts.edu) or to H.F.L. (email: lodish@wi.mit.edu)

VHHs are single-domain antibodies of molecular weight ~15 kD that are derived from the unusual heavy-chain-only antibodies produced by camelids[1]. Compared to conventional antibodies, VHHs are more stable and are typically better expressed in recombinant hosts. They also have a greater tendency to recognize conformational shapes (reviewed in ref. [2]). While single VHHs can be potent toxin-neutralizing agents, greatly improved therapeutic efficacy has been demonstrated in several animal models when two or more different toxin-neutralizing VHHs were linked and expressed as multi-specific VHH-based neutralizing agents (VNAs)[3–7]. Though VNAs are highly effective antitoxins in vivo, their half-life in circulation is relatively short[8], and it is thus important to improve the serum half-life of VNAs to substantially increase the duration of antitoxin protection.

We chose to use botulinum neurotoxin serotype A (BoNT/A) as our model toxin due to its importance as both a source of food poisoning and a potential bioweapon and the robust tools available for evaluating and quantifying antitoxin therapeutic efficacy. BoNT/A targets neurons and inhibits the release of neurotransmitters from presynaptic terminals by cleaving synaptosomal-associated protein of 25 kDa (SNAP25), a member of the soluble N-ethylmaleimide-sensitive factor-attachment protein receptor protein family[9]. Blocking neurotransmitter release at the neuromuscular junction leads to flaccid paralysis and death[10]. Because of its extreme potency and wide availability, BoNT/A is considered a category A bioweapon by the Centers for Diseases Control and Prevention[10]. We have previously reported development and testing of potent antitoxin VNAs for treating BoNT/A botulism[7, 11]. We also showed that appending an albumin-binding peptide to the C'-terminus of VHH anti-botulinum increases the serum half-life of this VHH to 1–2 days[11]. Other groups have also administered a combination of biotinylated VHH anti-BoNT/A and a fusion protein consisting of a scFv-specific to murine glycophorin A (GPA) and streptavidin. This strategy increases the VHH retention time in the circulation as well as its neutralization potency in mice. Nevertheless, neutralization protection only lasts for a maximum of 4 days, indicating the short-lived stability of the complex[12].

Aiming to improve the half-life of VHHs and VNAs in the circulation in vivo, we covalently linked these agents to proteins on the surface of red blood cells (RBCs), the most abundant cell type in the human body. Because RBCs have a circulatory half-life of 120 days in humans and ~30–50 days in mice, attached cargoes can potentially circulate for weeks unless modified RBCs are cleared more rapidly. RBCs also possess a natural biocompatibility and a large surface area, allowing covalent attachment of large numbers of cargoes without provoking adverse immune reactions[13–15]. Indeed, RBC transfusion has been carried out for centuries with negligible side effects. Lastly, RBCs lack nuclei, eliminating risks associated with administration of nucleated cells, especially those previously subjected to genetic manipulation[14, 16].

Several techniques have been developed to attach therapeutic cargoes to the surface of RBCs[17]. Covalent chemical attachment provides strong cargo binding but chemical medication methods are not specific and may alter membrane properties and blood antigens[18, 19]. Antibody-mediated binding to RBCs has also been employed, though cargo dissociate over time[20]. To preserve the native biological properties of RBCs and to seek to prolong the circulation time of VHHs, we developed a different strategy in which we generated virus vectors encoding chimeric proteins with one or more VHHs fused in frame to a cDNA encoding the RBC membrane proteins GPA or Kell. By expressing these cDNAs in murine RBC progenitors and generating RBCs either by in vitro culture or stem cell transplantation, we show

that the half-life of anti-BoNT VHHs can be extended to several weeks, equal to that of unmodified RBCs. These anti-BoNT VHHs are functional; they bind, neutralize, and remove BoNT/A from the circulation. Mice transfused with engineered RBCs accounting for <1% of total RBCs are resistant to multiple lethal doses of BoNT. Finally, we demonstrate that human RBCs can be produced in culture and engineered to express chimeric proteins containing anti-BoNT VHHs. Transfusion of these engineered RBCs into mice protects them from death caused by BoNT challenge. We suggest that similar types of engineered human RBCs can be used to provide long-term protection against exposures to a variety of bacterial toxins and harmful viruses.

## Results

**In vitro characterization of engineered mouse reticulocytes.** We first generated retroviral constructs that can infect red cell progenitors and lead to expression of chimeric proteins on the surface of mature RBCs. We chose GPA and Kell as the RBC membrane protein targets and genetically fused antitoxin VHHs at the N'-terminus of GPA and at the C'-terminus of Kell, respectively, to expose the VHHs on the external surface of the RBC plasma membrane (Fig. 1a). Both BoNT serotype A (BoNT/A) and serotype B (BoNT/B) are extremely toxic to humans; the two bispecific VNAs, termed VNA/A and VNA/B, recognize and neutralize BoNT/A and BoNT/B, respectively. VNA/A and VNA/B are heterodimers of two linked VHHs that recognize different epitopes on BoNT/A (ciA-H7 and ciA-B5)[3] or BoNT/B (JLU-D10 and JLI-G10, unpublished). The two VHHs are separated by a (GGGGS)$_3$ flexible spacer, and a myc-tag was added to the N'-terminus of GPA-VNA and the C'-terminus of Kell-VNA to simplify analysis. We infected E14.5 mouse fetal liver red cell progenitors with these retroviruses as well as with an empty vector as control. The progenitors were then cultured in vitro to differentiate into reticulocytes. Terminally differentiated cells express the chimeric GPA or Kell proteins on their surface, as indicated by myc surface expression (Supplementary Fig. 1a). Each RBC was estimated to express ~4,600,000 copies of GPA-VNA and 2,200,000 copies of Kell-VNA proteins per cell (Supplementary Fig. 1b; see Methods section and legend to Supplementary Fig. 1b for calculation). In this in vitro mouse fetal liver culture system, the VNA-expressing cells undergo enucleation at a level similar to control cells and have similar CD71 and Ter119 surface expression, proliferation, and morphology compared to control cells, suggesting that these modifications do not disturb normal red cell differentiation (Supplementary Figs. 1c–f).

We evaluated the ability of these engineered RBCs to neutralize BoNT/A using in vitro neutralization assays (Fig. 1b). Primary rat neurons were co-incubated with BoNT/A and various amounts of engineered RBCs. More than 70% of SNAP25 was cleaved in control cultures, while SNAP25 was mostly intact in neurons co-incubated with 1,000,000 or 100,000 GPA-VNA/A-expressing RBCs. Incubation with 1,000,000 or 100,000 Kell-VNA/A-expressing cells also protected neurons from BoNT/A. The protective ability of the GPA-VNA/A and Kell-VNA/A was specific since there was no protection conferred by RBCs with surface expressed VNA/B.

**Engineered mouse RBCs protect against BoNT/A in vivo.** Since our engineered RBC progenitors undergo normal erythropoiesis in vitro, we next produced engineered RBCs in vivo by transplanting irradiated mice with fetal liver red cell progenitors engineered to express GPA-VNA/A or Kell-VNA/A. Complete blood counts (CBCs) and myc surface expression (indicating the

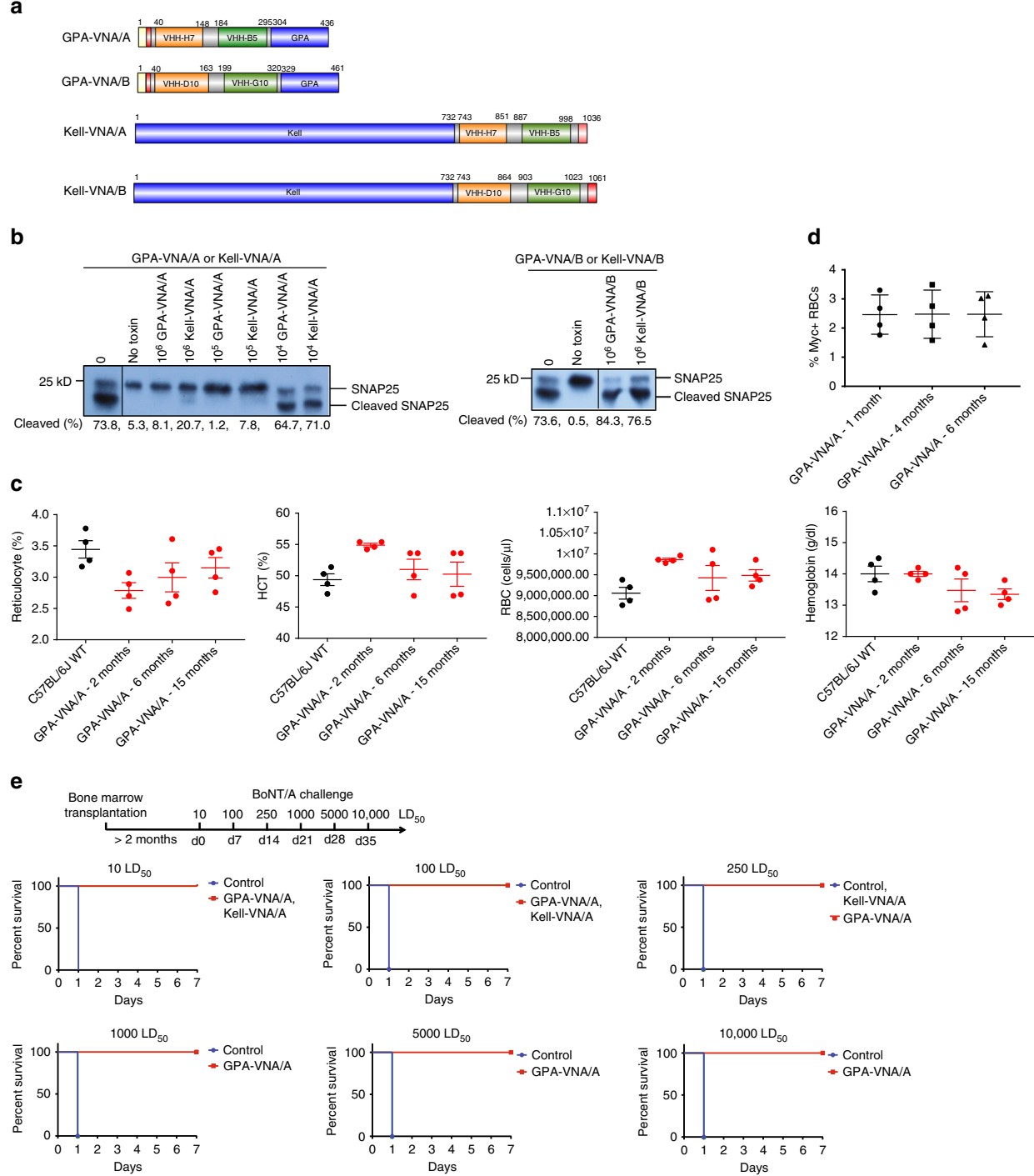

**Fig. 1** Genetically engineered murine RBCs covalently linked to VHHs against BoNT/A protect neurons in vitro and mice in vivo against BoNT/A challenge. **a** Design of the chimeric proteins. Each chimera was assembled from different protein segments shown as *colored boxes* (*yellow*: signal peptide of GPA; *red*: myc epitope; VHH-H7 and VHH-B5: anti-BoNT/A VHHs; *grey*: spacer; VHH-D10 and VHH-G10: anti-BoNT/B VHHs; GPA human glycophorin A). **b** RBC potency to neutralize BoNT/A assessed by SNAP25 immunoblot following overnight treatments of primary rat neurons exposed to 20 pM BoNT/A preincubated with the indicated number of myc+ RBCs. The percentage of SNAP25 cleaved by BoNT/A was estimated by image analysis. *Left*: RBCs expressing either GPA-VNA/A or Kell-VNA/A; *Right*: RBCs expressing GPA-VNA/B or Kell-VNA/B. **c** Complete blood counts of control 7-week-old female C57BL/6J mice and mice subjected to bone marrow transplantation with progenitor cells expressing vector or GPA-VNA/A and bled at the indicated time points. ($n = 4$/group, mean ± S.E.M.). **d** Myc surface expression on red cells from GPA-VNA/A transplanted mice was measured by flow cytometry at the indicated time points; the percentage of myc+ cells was determined. ($n = 4$/group, mean ± S.E.M.). **e** Kaplan–Meier survival plots of mice challenged with BoNT/A. CD1 mice were subjected to bone marrow transplantation with progenitor cells expressing GPA-VNA/A or Kell-VNA/A. After bone marrow reconstitution, these mice were challenged with 10 LD$_{50}$ BoNT/A and monitored for a week. The surviving mice received increasing doses of BoNT/A in subsequent weeks. The mice bearing Kell-VNA/A RBCs were challenged up to 3 weeks before protection faltered, whereas the mice carrying GPA-VNA/A were challenged for up to 6 weeks with increasing doses as indicated and were still alive at the termination of the study ($n = 5$/group). All mice with RBCs expressing GPA-VNA/A survived following the final 10,000 LD$_{50}$ treatment without showing signs of botulism

expression of the GPA-VNA/A chimera) from one batch of GPA-VNA/A-transplanted mice was followed over time (Fig. 1c, d). Compared to 7-week-old female mice, the blood parameters of the transplanted mice varied but were still within the normal range reported in the Mouse Phenome Database on the Jackson Laboratory website. Moreover, the average myc surface expression was stable over 6 months. In a separate batch of mice transplanted with cells expressing GPA-VNA/A, $2.96 \pm 0.10\%$ of the RBCs were myc+, and mice transplanted with cells expressing the Kell-VNA/A chimera had $14.26 \pm 0.25\%$ myc+ RBCs after 6 week bone marrow reconstitution ($\pm$S.E.M.; $n = 5$). As shown in Supplementary Fig. 1g, each of these RBCs contained about 310,000 GPA-VNA/A proteins per cell, about 1/15th that found on red cells made in our in vitro culture.

These mice were then challenged with BoNT/A. As shown in Fig. 1e, both GPA-VNA/A- and Kell-VNA/A-producing mice, but not wild-type mice, survived a 10 $LD_{50}$ BoNT/A challenge. Seven days later, we challenged these surviving mice with a 100 $LD_{50}$ of BoNT/A; both GPA-VNA/A and Kell-VNA/A mice survived. We then kept challenging the mice at weekly intervals with higher doses of BoNT/A. The Kell-VNA/A-expressing mice died at 250 $LD_{50}$; remarkably, the GPA-VNA/A mice survived even when challenged with 10,000 $LD_{50}$ BoNT/A.

The protective capacity of the engineered RBCs is versatile, since 60% of mice with $4.77 \pm 1.18\%$ ($\pm$S.E.M.; $n = 5$) GPA-VNA/B-expressing RBCs survived a BoNT/B challenge as high as 1,000 $LD_{50}$ (Supplementary Fig. 2) and 100% of mice expressing the GPA-VNA/B chimera survived a 100 $LD_{50}$ BoNT/B challenge.

Since red cells expressing the GPA-VNA/A chimeras seem to provide more potent protection in vivo than do Kell-VNA/A mice, we carried out our subsequent experiments utilizing GPA as the RBC membrane anchor proteins for our chimeric proteins.

It is likely that this potent protection stems from the continuous replenishment of engineered RBCs by the bone marrow of the reconstituted mice. Transfusion of engineered RBCs is a more realistic option to exploit engineered RBCs therapeutically in humans. Therefore, we transfused wild-type mice with 100 μl of blood drawn from chimeric mice in which $6.22 \pm 0.25\%$ of the RBCs displayed surface GPA-VNA/A ($\pm$S.E.M.; $n = 5$). As there are $10^7$ cells/μl of whole blood, we therefore injected $6.22 \times 10^7$ GPA-VNA/A-expressing RBCs. We then challenged the recipient mice with BoNT/A (Fig. 2a). Blood containing GPA-VNA/A RBCs protected mice from 10 $LD_{50}$ BoNT/A, although these mice died within 24 h if challenged with $\geq$100 $LD_{50}$ doses of BoNT/A. We reasoned that there were not enough cells to neutralize the higher doses of BoNT/A at the site where the toxins enter the circulation. We thus injected 400 μl of blood from the transplanted GPA-VNA/A chimeric mice into other recipients and found that these mice were protected from 100 $LD_{50}$ BoNT/A challenge (Fig. 2b).

Several reports showed that the half-life of VHHs in vivo is very short[21] and unmodified anti-BoNT/A VNAs protect mice from BoNT/A challenge for less than a day postadministration[11]. The half-life of an antibody can be prolonged, as shown in a human study that humanized monoclonal anti-BoNT/A antibodies had half-lives varying from 2.5 to 26.9 days depending on the dose of antibody administrated[22]. Here we analyzed the half-life of the transfused engineered RBCs in the circulation. To this end, we stained the RBCs with a violet-trace dye prior to transfusion to monitor the total population of transfused RBCs and tracked cells expressing VNAs by green fluorescent protein (GFP) signal (from a GFP expression cassette in the lentivirus vector) (Fig. 2c). The half-life of control, GPA-VNA/A, and Kell-VNA/A RBCs were all approximately 14 days, similar to that of normal mouse RBCs. We then challenged these mice with

10 $LD_{50}$ BoNT/A at different times after transfusion and found that the antitoxin-protective capacity of the engineered RBCs lasted up to 28 days (Fig. 2d). Taken together, our work demonstrated that covalently conjugating VNAs onto RBCs allows a dramatic extension of the circulatory half-life of the VNAs without compromising their neutralizing capacity.

**Engineered RBCs pose no detrimental side effects in vivo.** To determine the fate of BoNT/A and the engineered RBCs in mice, we incubated violet-dyed wild-type RBCs or GPA-VNA/A-expressing RBCs with catalytically inactive BoNT/A (ciBoNT/A) before transfusing the mixture into mice. We bled the recipient mice at intervals and analyzed amounts of both free (Fig. 3a) and RBC-bound ciBoNT/A (Fig. 3b) in the serum. As shown in Figs. 3a, 1 h after injection ("0 day" on the graph) there is very little free ciBoNT/A in the serum of mice transfused with GPA-VNA/A-expressing RBCs, compared to mice transfused with control RBCs. In contrast, ~100 times more RBCs appear to have bound ciBoNT/A in mice transfused with GPA-VNA/A-expressing RBCs than in mice transfused with control blood based on the RBC surface S-tag signal (from the anti-ciBoNT/A probe; Fig. 3b). RBC-bound ciBoNT/A is cleared from serum after 7 days (Fig. 3b), approximately concurrent with the GPA-VNA/A (GFP+ cells) (Fig. 3c). Since the GPA-VNA/A-expressing cells (GFP+ cells) were cleared significantly faster than the other RBCs (Fig. 3c) and similar to bound ciBoNT/A (Fig. 3b), a high percentage of GPA-VNA/A RBCs must have become bound to ciBoNT/A. We hypothesize that binding of BoNT/A to GPA-VNA/A-expressing RBCs enhances their degradation by macrophages or dendritic cells in the spleen or liver[23–25].

We performed the same experiment but bleeding intervals were taken within 1 h after transfusion. Transfused GPA-VNA/A RBCs still bound ~100 times more ciBoNT/A than control RBCs, despite the ~20% loss of bound ciBoNT/A from the GPA-VNA/A RBCs at 1 h compared to 5 min (Supplementary Fig. 3). Since undetectable numbers of transfused cells were lost between 5 min and 1 h, it was appropriate to use 1 h as "time 0" in the experiment in Fig. 3a–c.

Multiple repeat antibody administration typically elicits an adverse immune response. We therefore investigated the immune response in mice after three injections of control RBCs, GPA-VNA/A RBCs, or equimolar amounts of recombinant VNA/A (Fig. 3d). We used enzyme-linked immunosorbent assay (ELISA) to detect the relative abundance of anti-VNA/A antibodies in the serum. At 1:333 and 1:111 serum dilutions, very little antibody was detected in mice receiving control or GPA-VNA/A RBCs, whereas large amounts of anti-VNA/A antibody were detected following injection of the VNA protein. As shown by the ELISA performed at a 1:37 serum dilution, the amount of anti-VNA produced against the VNA-expressing red cells is at least 10-fold less than that from VNA/A-injected mice.

**Heterodimeric VHHs protect more robustly than monomeric VHHs.** Given that murine RBCs expressing the GPA-VNA/A chimera are very potent, we also examined whether monomeric VHHs can be used instead of bispecific VNAs. Thus we constructed another plasmid (GPA-VHH7) to generate a chimeric protein fusing ciA-H7 VHH with GPA (Fig. 4a). We then produced cultured mouse RBCs expressing GPA-VNA/A and GPA-VHH7 and showed that they express similar levels of chimeric GPA and have similar BoNT/A-neutralizing potency in neuronal cell-based assays (Fig. 4b). Mice transfused with an equivalent amount of cultured RBCs expressing either GPA-VNA/A or GPA-VHH7 were protected from 25 $LD_{50}$ of

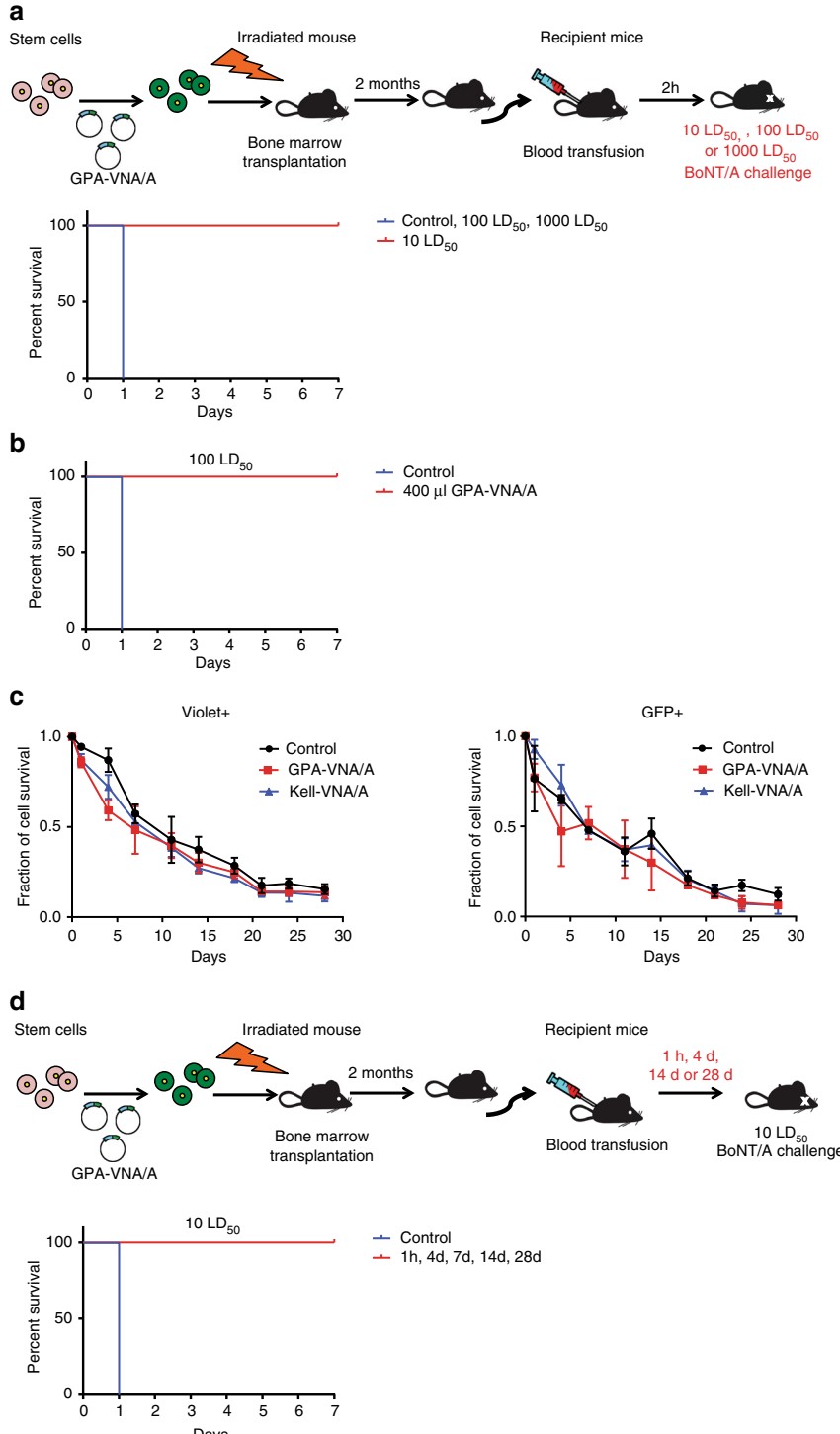

**Fig. 2** Mice transfused with engineered RBCs are protected against BoNT/A challenges. **a** C57BL/6J mice were transfused with 100 μl blood from chimeric mice previously transplanted with GPA-VNA/A-expressing progenitors. The recipient mice were challenged with 10, 100, or 1000 LD$_{50}$ BoNT/A 2 h later and survival was monitored ($n = 6$/group). Note: the curves depicting control mice and transfused mice challenged with 100 and 1,000 LD$_{50}$ are overlapping. **b** C57BL/6J mice transfused with 400 μl blood from chimeric mice previously transplanted with GPA-VNA/A-expressing progenitors were challenged with 100 LD$_{50}$ BoNT/A 1 h after transfusion and survival was monitored ($n = 4$/group). **c** Circulatory half-life of the transfused RBCs. In all, 100 μl of blood from transplanted mice with ~3% of their RBCs expressing vector only, GPA-VNA/A, or Kell-VNA/A were stained with violet-trace dye and transfused into recipient mice. The fraction of transfused RBCs in recipients was analyzed by flow cytometry at the indicated time points. The violet-trace dye represents the total population of transfused red blood cells, of which only ~3% are GFP+ and express the exogenous chimeric protein, while the GFP signal represents only the 3% of the transfused RBCs expressing the VNAs. ($n = 3$/group, mean ± S.E.M.). **d** Survival plot of transfusion recipient mice treated as in the diagram were challenged with 10 LD$_{50}$ BoNT/A at 1 h, 4 days, 14 days and 28 days post-transfusion and monitored for 7 days ($n = 6$/group)

BoNT/A (Fig. 4c) while only the GPA-VNA/A mice were protected from 50 LD$_{50}$, and to a lesser extent at even higher doses. The results suggest that there may be a small potency advantage in vivo when using RBCs displaying bispecific VNAs vs. monospecific VHHs.

**Engineered human reticulocytes expressing heterodimeric VHHs.** Finally, we explored the possibility of producing engineered human RBCs from human CD34+ progenitor cells. We modified our current CD34+ cell culture protocol and measured the cell number, differentiation markers, membrane proteins, and enucleation at the end of each developmental stage. As shown in Table 1, we developed a six-stage culture system, a modification of the four-stage system used previously in our laboratory and the three-stage system used by other laboratories[26], [27]. In our modified system, mobilized bone marrow CD34+ stem/ progenitor cells expanded >300,000-fold during the 23-day culture period (Fig. 5a). Differentiation was gradual and synchronous; as an example, c-kit expression was initially high and declined with differentiation, whereas CD71 and CD235A expression increased. At the time when CD71 (transferrin receptor) expression began to decrease, cells began to enucleate; >90% of cells had undergone enucleation and became reticulocytes by the end of the culture (Fig. 5b). The enucleated cells are 7 µm in diameter, similar to human reticulocytes[28], and contain 28.23 ± 0.46 pg hemoglobin/cell ($n = 3$) within the normal range for human RBCs (Fig. 5c)[29]. We also monitored the survival of these in vitro-differentiated human reticulocytes in vivo and found that they can circulate for at least 7 days in macrophage-depleted nonobese diabetes/severe combined immunodeficiency (NOD/SCID) mice (Fig. 5d). About 94% of cultured reticulocytes survived in these mice at 1 day post-transfusion and ~36% cells at 3 days, comparable to the recent published data[30]. Overall, we improved the system in terms of the number of differentiated, enucleated cells (93.53 ± 2.84% enucleation; $n = 3$). We compared the expression of multiple surface proteins at the end of each culture stage and showed that the expression pattern of these proteins is similar to that previously reported[31]. As examples, GPA (CD235A) and Rhesus (CD240DCE) increase over time, and CD36, CD47, CD59, CD71, CD147, α4 integrin (CD49d), α5 integrin (CD49e), β1 integrin (CD29), and Kell show decreased expression during differentiation (Fig. 6a).

We then infected CD34+ stem/progenitor cells with lentiviral vectors that express the hGPA-VNA/A or hKell-VNA/A chimeric proteins depicted in Fig. 1a. Importantly, control and GPA-VNA/A-expressing cells show similar expression patterns of multiple surface proteins, suggesting that expression of this chimeric protein does not significantly alter differentiation (Fig. 6a). Western blotting analysis showed that our six-stage cultured control or GPA-VNA/A-expressing reticulocytes contain similar amounts of GPA, Kell, β1-integrin, band 3, and XK proteins, compared to human RBCs (Supplementary Fig. 4b). Moreover, the morphology of vector control and GPA-VNA/A-expressing CD34+ cells are similar to uninfected CD34+ cells at each stage, indicating that the cells differentiate properly (Fig. 6b). However, infection of CD34+ cells leads to a reduced number of cells undergoing enucleation (to ~70–80%; Fig. 6b). The growth rate is unperturbed following the expression of GPA-VNA/A (Fig. 6c).

Enucleated cells express ~500,000 copies of hGPA-VNA/A or 120,000 copies of Kell-VNA/A, as indicated by myc expression (Supplementary Fig. 4a, see legend for calculation). The number of chimeric proteins in engineered human cells is lower than that in engineered mouse cells, which could result from varying numbers of viral particles infecting cells and from the use of different vectors with different promoters.

**Engineered human reticulocytes protect against BoNT/A.** We co-incubated these engineered human RBCs with neurons in the

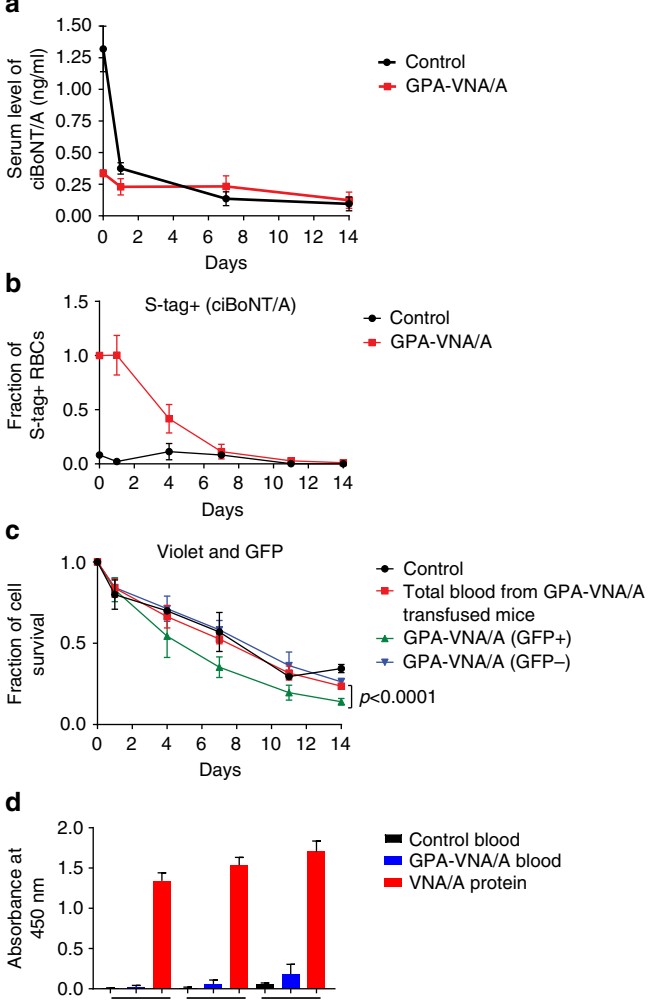

**Fig. 3** No detrimental side effects in mice injected with engineered RBCs. **a** Detection of serum persistence of unbound ciBoNT/A in the circulation. In all, 2 ng ciBoNT/A was incubated with 200 µl violet-stained RBCs expressing the control vector or GPA-VNA/A before they were transfused into recipient mice. The sera was collected at intervals and the amount of unbound ciBoNT/A in the serum was measured by ELISA ($n = 3$/group, mean ± S.E.M.). **b**, **c** Detection of RBC-bound ciBoNT/A and transfused RBCs in the blood of recipient mice. In all, 200 µl blood from wild-type mice and from mice transplanted with GPA-VNA/A expressing RBCs was stained with violet-trace dye and incubated with 1 µg ciBoNT/A before transfusion into mice. Recipients were bled at the indicated time points. RBCs were subjected to flow cytometric analyses to quantify the violet trace (total transfused RBCs, panel **c**), GFP (virus-transduced cells, **b**), and S-tag (indirectly detecting RBC-bound ciBoNT/A, panel **b**) ($n = 3$/group, mean ± S.E.M.). In Fig. 2c, the GPA-VNA/A RBCs measures total RBCs from GPA-VNA/A chimera mice, which are a combination of non-transduced (~97%) and transduced cells (~3%). To further distinguish the transduced cells, we gated on the GFP+ (GPA-VNA/A) RBCs and GFP– populations (control RBCs). The difference between the GFP+ and GFP– curves is statistically significant by ANOVA two-tailed analysis. **d** As detailed in the Methods section, mice received three injections of control blood, GPA-VNA/A blood, or VNA/A protein, and relative abundance of antibody against-VNA/A in serum from these mice was examined by ELISA. Sera were diluted at the indicated ratios ($n = 5$/group, mean + S.E.M.)

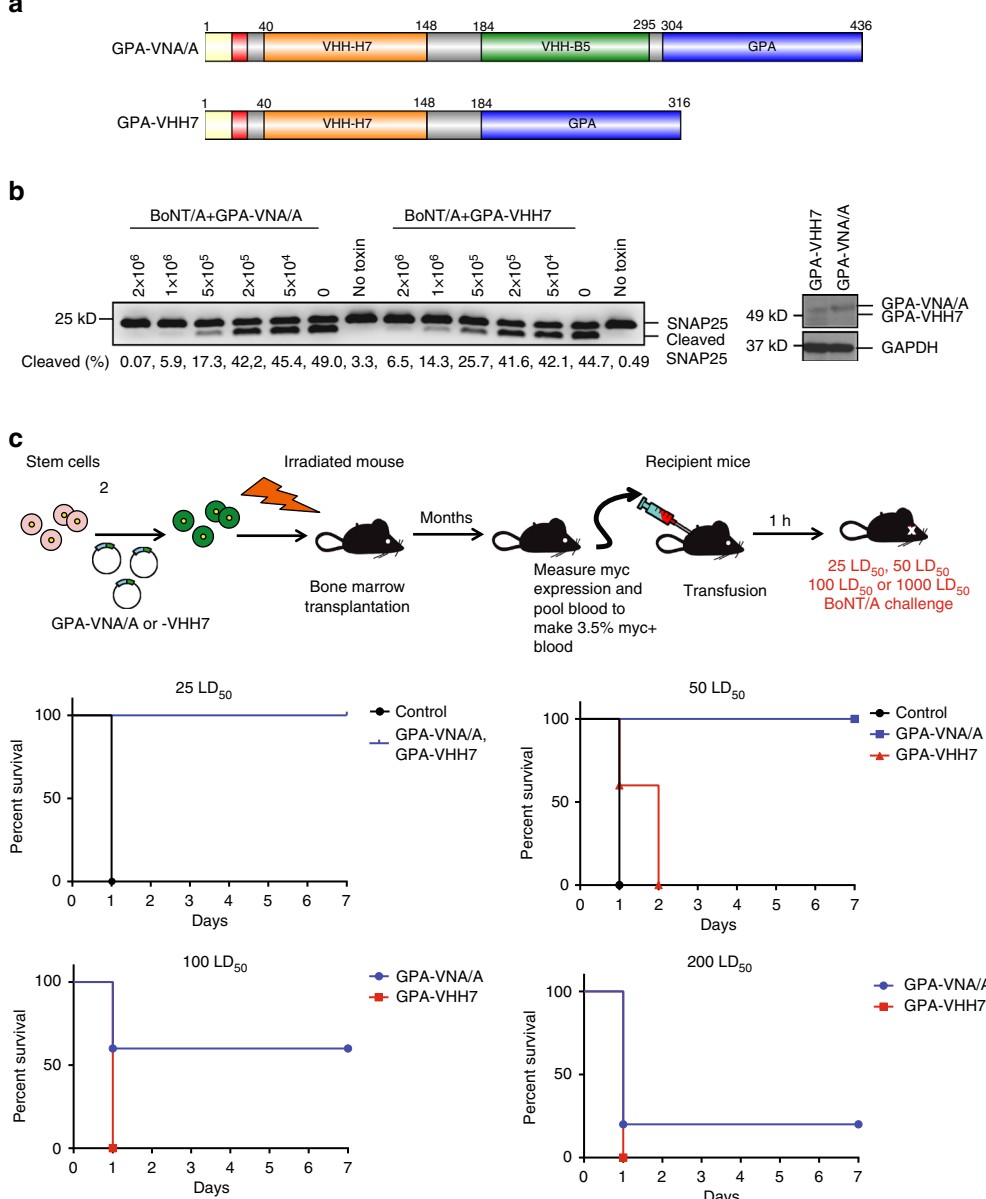

**Fig. 4** RBCs expressing heterodimers of neutralizing VHHs are more protective than those expressing monomers. **a** Bispecific or monospecific antitoxin proteins were engineered for expression on RBCs as GPA fusions. Chimeras were engineered to include the different protein segments as shown (*yellow* signal peptide of human glycophorin A; *red* myc epitope; *grey* spacer). **b** RBC potency to neutralize BoNT/A assessed by SNAP25 immunoblot following overnight treatments of primary rat neurons exposed to 20 pM BoNT/A preincubated with the indicated number of myc+ RBCs. The percentage of SNAP25 cleaved by BoNT/A was estimated by image analysis and shown below the immunoblots. **c** Survival plot of transfusion recipient mice challenged with BoNT/A. C57BL/6J mice were transfused with 100 µl blood from chimeric mice with blood containing 3.5% RBCs expressing either GPA-VNA/A or GPA-VHH7. Mice were then challenged with 25, 50, 100, or 200 LD$_{50}$ BoNT/A and monitored for 7 days ($n = 5$/group)

presence of BoNT/A. Co-incubation with one million hGPA-VNA/A or hKell-VNA/A-expressing human RBCs partially inhibited BoNT/A activity and five million hGPA-VNA/A-expressing human RBCs fully prevented SNAP25 cleavage (Fig. 6d). Our in vitro-cultured human reticulocytes survive following transfusion in NOD/SCID mice for at least 7 days (Fig. 5d) and this ability allowed us to examine the protective capacity of the engineered human RBCs in vivo. Furthermore, the percentage of surviving, circulating reticulocytes expressing vectors or GPA-VNA/A, as well as mature human RBCs, in NOD/SCID mice is similar from 5 min post-transfusion to 1 h (Supplementary Fig. 4c). We therefore transfused 150,000,000 hGPA-VNA/A-expressing human reticulocytes into NOD/SCID mice and challenged them with 10 LD$_{50}$ of BoNT/A

30 min-post-transfusion. Consistent with the in vitro toxin-neutralization results, these human reticulocytes provided protection against 10 LD$_{50}$ BoNT/A in all recipient mice (Fig. 6e). Since our hGPA-VNA/A-expressing human reticulocytes showed similar morphology and vitality compared to its unmodified in vitro-cultured human reticulocytes counterpart, we expected that >90% of GPA-VNA/A cells would survive 1 day post-transfusion in NOD/SCID mice and protect mice from a low-dose challenge with BoNT/A. Indeed, when we transfused a cohort of NOD/SCID recipient with 120,000,000 hGPA-VNA/A-expressing human reticulocytes and challenged them 1 day post-transfusion with 10 LD$_{50}$ of BoNT/A, these mice were protected (Fig. 6e).

**Table 1 Composition of the culture medium at each stage**

| Stage | Expansion | Diff1 | Diff2 | Diff3 | Diff4 | Diff5 |
|---|---|---|---|---|---|---|
| Day | D1–D5 | D5–D9 | D9–D12 | D12–D16 | D16–D20 | D20–D23 |
| Density | 1E5/ml | 1E5/ml | Dilute into 4X | 1E5/ml | 1E6/ml | 5E6/ml |
| Medium | Stemspan | IMDM | IMDM | IMDM | IMDM | IMDM |
| | rhFLT3 | Plasma | Plasma | Plasma | Plasma | Plasma |
| | rhSCF | AB serum | AB serum | AB serum | AB serum | AB serum |
| | rhIL-3 | Heparin | Heparin | Heparin | Heparin | Heparin |
| | rhIL-6 | Insulin | Insulin | Insulin | Insulin | Insulin |
| | Dex | Holo-transferrin | Holo-transferrin | Holo-transferrin | Holo-transferrin | Holo-transferrin |
| | | SCF | SCF | SCF | Epo | |
| | | IL-3 | IL-3 | Epo | | |
| | | Epo | Epo | | | |

## Discussion

Here we describe the production of genetically engineered RBCs that carry cargoes of therapeutic value: in this case, VHHs that neutralize a bacterial toxin and that have long circulatory half-life. First, we demonstrated that VHHs fused to a GPA or Kell protein can be functionally expressed on the surface of normal enucleated mouse RBCs and human reticulocytes. These modifications of RBCs do not materially affect their biogenesis, as judged by surface protein expression, proliferation, cell size, and hemoglobin content. Second, these chimeric VHHs retain their potency in in vitro BoNT/A-neutralization assays and in vivo toxin challenges. Notably, mice transfused with RBCs expressing the GPA-VNA/A chimera such that the engineered cells comprise <1% of the total mouse RBCs survive a BoNT/A challenge of 10 LD$_{50}$ BoNT/A for up to 28 days. Third, we used a modified culture system and genetically modified human CD34+ stem/progenitor RBCs to produce enucleated human reticulocytes expressing the hGPA-VNA/A chimera and showed that transfusion of these cells into immune-compromised mice rendered them resistant to challenge by a 10 LD$_{50}$ dose of BoNT/A. Our RBC engineering method should have widespread applications for prolonging half-life in the circulation of any enzyme or anti-toxin used for prophylactic or therapeutic treatments.

We previously reported sortase A-mediated modification (sortagging) to attach cargoes covalently to RBCs[32]. Sortase A recognizes an LPXTG motif and cleaves the peptide bond between the threonine and glycine residues to yield a thioester acyl-enzyme intermediate. A nucleophile that contains suitably exposed N-terminal glycines, $(G)_n$, can resolve this intermediate, covalently linking the two motifs via a peptide bond[33]. Despite the flexibility of cargo selection provided by sortase A-mediated cargo delivery, human and mouse red cells contain only ~3,000–8,000 surface proteins with N- terminal $(G)_n$ motifs[34], which limits the cargo-loading numbers. The genetic engineering method detailed in this report provides a way to bypass this challenge, permitting greatly increased cargo capacity.

Compared with other RBC engineering methods, our methods are better suited for long-term, persistent delivery of cargo. For instance, RBC membrane-coating techniques produce RBC-membrane-camouflaged polymeric nanoparticles by deriving membrane vesicles from RBCs and fusing these vesicles with nanoparticles. This protocol enables the cargo to last ~50 h in circulation[35], while our genetically engineered mouse RBCs circulate in the bloodstream for ~28 days. Covalent attachment of cargo onto RBCs not only prolongs in vivo retention times of chimeric proteins but also avoids their rapid clearance[8]. Interestingly, we observed that the engineered RBCs that have bound the antigen (toxin in our experiments) are cleared slightly faster than are unperturbed engineered RBCs. It is not clear whether this half-life difference is due to the large size of the bound BoNT/A (150 kDa) or the binding of antigen itself; it will be interesting to attach other VHHs, whose target antigens differ in size and other properties, and determine the effects on RBC clearance. Another possibility is that these toxin-carrying RBCs are somehow seen by the cells of the reticuloendothelial system as damaged RBCs and cleared by macrophages or dendritic cells.

We showed that a single VHH (GPA-VHH7) is also able to neutralize BoNT/A. Since each VHH comprises a single immunoglobulin domain stabilized by one or two intramolecular disulfide bonds that fold independently[36, 37], it is likely possible to engineer GPA or Kell chimeras that contain three or more VHH domains and express these on the same RBCs. In this way, one could engineer RBCs that bind multiple foreign toxins or viruses and thus offer long-term prophylactic protection against multiple pathogens.

Systemic administration of foreign proteins carries significant risk of inducing a strong antibody response. This is especially the case for neutralizing antibodies derived from mice[38]. Such agents usually require humanization at the expense of reducing affinity and specificity. In worst case scenarios, even with humanization, they may prove too immunogenic to be administered more than a handful of times[39]. Here we assessed whether long-term protection against BoNT/A comes with a side effect immune response by carrying out repeated transfusions of GPA-VNA/A RBCs into a cohort of C57BL/6J recipients at days 1, 21, and 28. In parallel, we also transfused unmodified RBCs or equimolar amounts of recombinant VNA/A proteins with the same schedule. Multiple transfusions of GPA-VNA/A RBCs were found to elicit an insignificant antibody response against intact VNA/A proteins, comparable to mice receiving unmodified RBCs, whereas mice that received VNA/A proteins developed a much higher titer of antibody response against intact VNA/A proteins (Fig. 3d). This result implies that the RBC vehicle, possibly by masking the VNA as the RBC's own proteins, can further minimize immune response against the covalently attached VNA. Recently, we showed that red cells with several different peptides covalently linked to their surface do not induce an immune response to these peptides after transfusion into a naive mouse. Rather, they induce tolerance to these peptide antigens[34]. Therefore, GPA-VNA/A RBCs and other engineered RBCs generated using similar methods can not only provide long-term protection but may also permit multiple administrations without provoking the immune system.

Based on previously reported culture systems using mobilized human bone marrow CD34+ cells to produce RBCs, which yield a lower number of cells[26] and a lower percentage of enucleated cells[30, 31, 40], our six-stage culture system offers enhanced enucleation. We increase the cell seeding density and reduce the erythropoietin level during the last stage, suggesting that these two factors positively affect reticulocyte production. Despite the higher

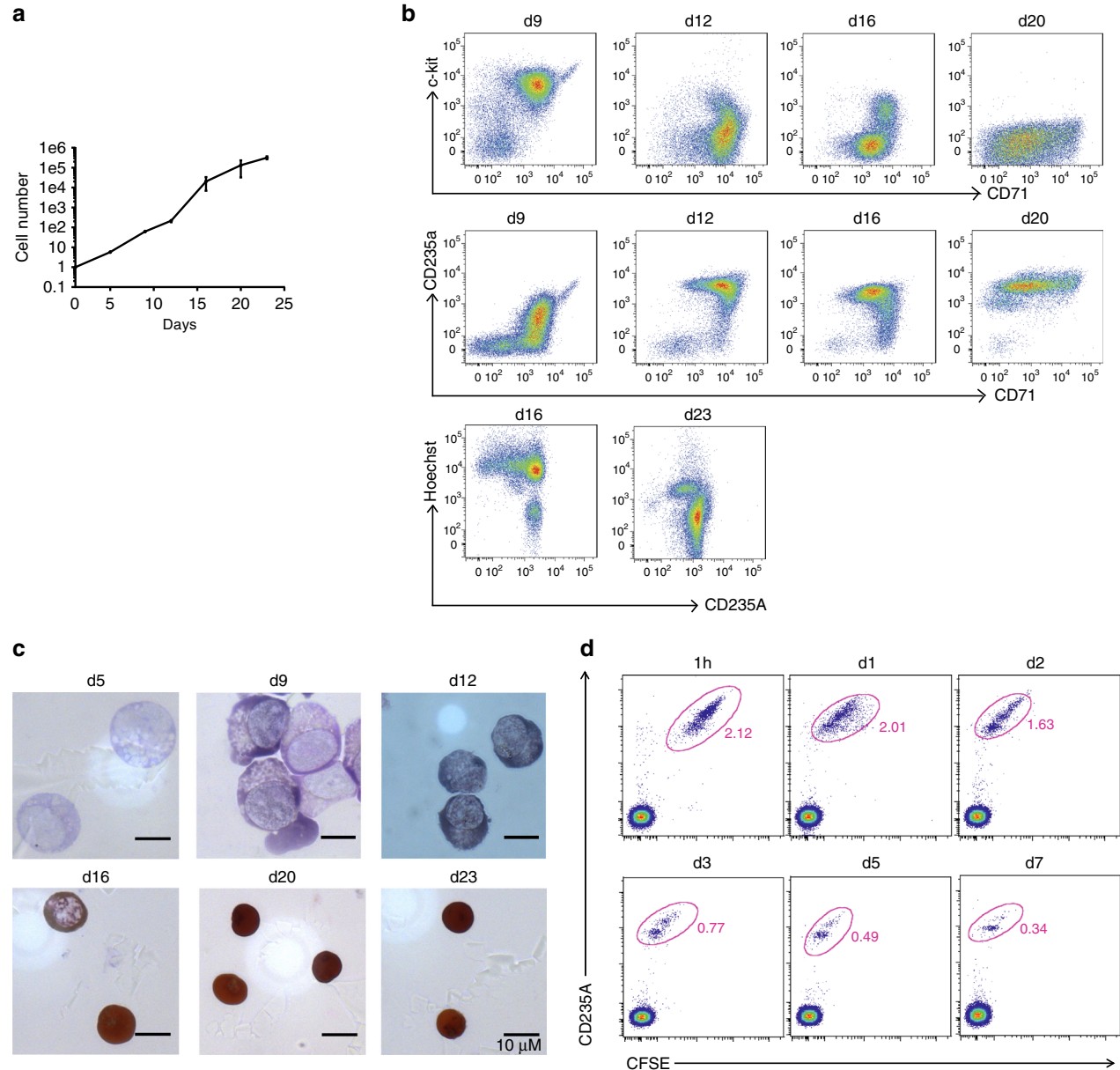

**Fig. 5** Characterization of in vitro-differentiated human RBCs. **a** The proliferation curve of human RBCs. ($n = 3$, mean ± S.E.M.). **b** Differentiating cells were characterized by flow cytometry for c-kit, CD71, CD235A, and Hoechst at the end of each culture stage. **c** Giemsa and hemoglobin staining of human RBCs at the end of each culture stage. **d** In vitro-differentiated human RBCs circulate for up to 7 days in macrophage-depleted NOD/SCID mice. Five hundred million six-stage cultured RBCs were labeled with CFSE and injected intravenously into NOD/SCID mice that have been treated with clodronate liposomes. The recipient mice were then bled at the indicated time points as indicated for further flow cytometric analyses. The transfused human RBCs were identified by tracing CFSE and CD235A expression

percentage of enucleated cells, we have not investigated the maturation status of the reticulocytes and cannot conclude that these cells are fully mature red cells. Reduced enucleation is observed in cells infected with lentiviruses, raising the safety concern of transmission of genetic material. While one possible solution is to filter nuclei and nucleated cells by leucocyte filtration as performed in other studies[29], adjusting the transduction protocol should be another possible method to reduce influence on culture efficiency. On the other hand, there are several clinical trials of cellular therapy using lentivirus-transduced cells that show great safety[41, 42]. Improving vector designs to enable safe delivery should be a consideration when engineering human RBCs in vitro.

Previous reports showed that the enucleated reticulocytes produced in culture mature normally into biconcave mature RBCs following transfusion into NOD/SCID mice, eliminating the necessity of further in vitro maturation[43]. More importantly, Giarratana et al.[43] have shown that the half-life of in vitro-cultured RBCs in the human body is ~26 days. If this half-life is confirmed in further studies, our engineered human reticulocytes carrying BoNT/A VHHs, which are produced in vitro, should provide effective long-term protection from lethal BoNT/A challenges in humans.

## Methods

**Plasmids and generation of recombinant retroviruses and lentiviruses**. Murine stem cell virus (MSCV)-based retroviruses were produced and used to infect erythroid progenitors following a previously described protocol[44], and lentivirus vectors used to infect human cells were also described previously[45]. The ciA-H7

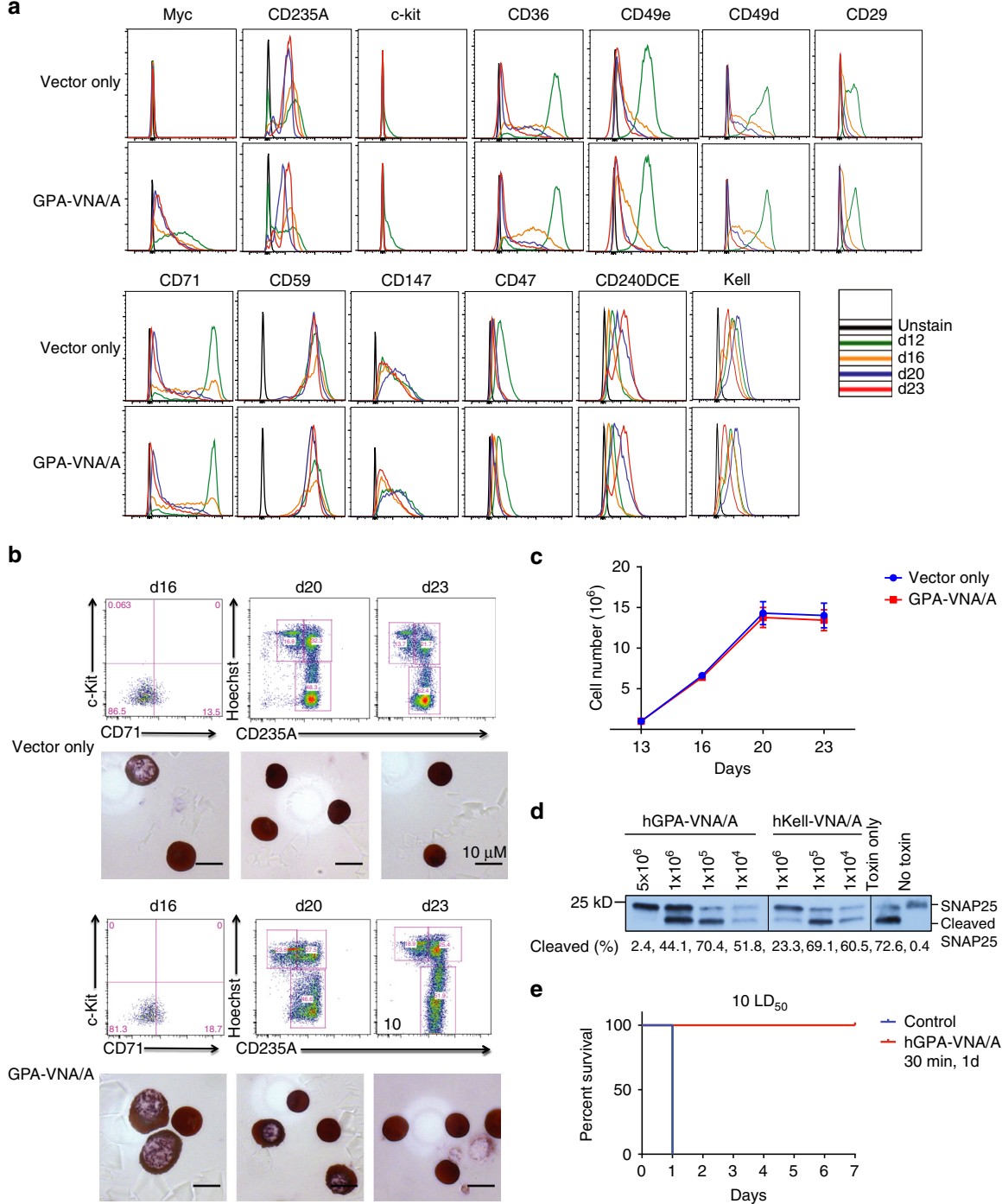

**Fig. 6** Genetically engineered human RBCs made in culture and expressing GPA-VNA/A or Kell-VNA/A differentiate normally and protect neurons against BoNT/A challenge both in neuronal culture and in vivo. **a** Mobilized human CD34+ cells infected with lentiviruses expressing the chimeric GPA-VNA/A protein were cultured by the method detailed in Fig. 4 and the expression of multiple surface proteins was examined by flow cytometry at the indicated time points. **b** *Upper panel* shows CD235A and Hoechst staining of human cells expressing GPA-VNA/A generated from CD34+ cells that have been cultured in vitro for 20 and 23 days. *Lower panel* shows Giemsa and hemoglobin staining of hRBCs expressing GPA-VNA/A at d20 and d23. **c** Proliferation curve during culture of mobilized human CD34+ cells expressing vector or GPA-VNA/A. ($n = 3$/group, mean ± S.E.M.). **d** Human RBCs expressing GPA-VNA/A or Kell-VNA/A protect cultured neuronal cells from BoNT/A protease activity. Rat neurons were co-incubated with BoNT/A and engineered hRBCs as indicated and neuronal lysates were analyzed by western blotting as in Fig. 1. **e** Survival curve of mice challenged with BoNT/A. Human RBCs generated by in vitro culture were submitted to flow cytometry to detect the percentage of GFP+ cells before injecting into mice. One hundred and fifty million GFP+ human RBCs expressing empty vector or GPA-VNA/A were transfused into NOD/SCID mice. After 30 min, the mice were challenged with 10 $LD_{50}$ BoNT/A and were observed for 7 days ($n = 4$/group). Mice were challenged with 10 $LD_{50}$ BoNT/A 1 day after injection of 120 million GFP+ human RBCs and was observed for 7 days ($n = 2$/group)

and ciA-B5 monomer sequences and the bispecific H7/B5 heterodimer sequence were previously reported[3] and the D10 and G10 sequences are unpublished (but are available from Dr. Shoemaker). Human GPA and Kell cDNA sequences were obtained from the National Center for Biotechnology Information, reference sequences NM_002099.6 and NM_000420.2, respectively. These sequences were used for synthesizing DNA fragments; these DNA fragments were cloned into XhoI-cut XZ201 or HMD plasmids using a Gibson Assembly Kit. HMD vector uses a HIV/MSCV hybrid long terminal repeat as the promoter. The XZ201 plasmid was used for retrovirus production and the HMD plasmid was used for lentivirus production. Both contain GFP inserted after an IRES sequence, which drives the GFP expression.

293T cells were cultured in Dulbecco's Modified Eagle Medium (DMEM) with 10% fetal bovine serum (FBS) in a humidified 5% $CO_2$ atmosphere at 37 °C. XZ201-based plasmids and pCLECO- or HMD-based plasmids and packaging vectors, VSV-G and pD8.9, were incubated with medium and Fugene 6 according to the Promega protocol. The mixture medium was changed after 6 h; retrovirus and lentivirus were collected after 24 and 72 h for transfection, respectively.

**Flow cytometric analyses and antibodies.** All flow cytometric data were acquired on a fluorescence-activated cell sorter (FACS) Fortessa flow cytometer (BD Biosciences) and analyzed using the Flowjo software (Tree Star). All stainings were carried out in FACS buffer (2 mM EDTA and 5% FBS in phosphate-buffered saline (PBS)) for 40 min at room temperature unless otherwise described. Samples were washed twice with FACS buffer prior to flow analyses. The following are the antibodies used at 1:100 dilution: anti-human CD235A-APC (eBioscience, 17-9987042), anti-human CD71-FITC (eBioscience, 11-0719-42), anti-human CD71-PeCy7 (Affymetrix, 25-0719-42), anti-human CD117-PeCy7 (eBioscience, 25-1178-42), anti-human CD117-BV605 (BioLegend, 313217), anti-human CD49e-APC (BioLegend, 328012), anti-human CD29-PerCP-eFluor 710 (Affymetrix, 46-0299-41), anti-human CD49d-PE (Affymetrix, 12-0499-42), anti-human CD240DCE-APC (Miltenyi Biotec, 130-104-818), anti-human CD238-APC (Miltenyi Biotec, 130-104-951), anti-human CD47-PerCP-eFluor 710 (Affymetrix, 46-0479-42), anti-human CD147-PE(Affymetrix, 12-1472-42), anti-human CD59-APC (Affymetrix, 17-0596-42), anti-myc tag-PE (Cell Signaling Technology, 3739), anti-mouse Ter119-APC (eBioscience, 17-5921-83), and anti-mouse CD71-PE (Affymetrix, 12-0711-83). Hoechst 33342 (Life Technologies, H1399) was used to visualize nuclei.

**Human CD34+ cell culture.** Granulocyte-colony stimulating factor-mobilized CD34+ peripheral blood stem cells (purchased from the Fred Hutchinson Cancer Center) were thawed according to the vendor's protocol. Cells were then placed in expansion medium containing 100 ng/ml rhFLT3, 100 ng/ml recombinant human stem cell factor (rhSCF), 20 ng/ml rhIL6, 20 ng/ml rhIL3, and 100 nM dexamethasone in Stemspan II medium for 5 days at a density of 100,000 cells/ml. The cells were then cultured in differentiation 1–2 medium (2% human blood plasma, 3% human serum, 3 U/ml heparin, 10 µg/ml insulin, 200 µg/ml holo-transferrin, 10 ng/ml rhSCF, 1 ng/ml interleukin-3, and 3 U/ml Epo in Iscove's Modified Dulbecco's Medium (IMDM)) at a density of 100,000 cells/ml for 4 days and at 200,000 cells/ml for 3 days. The medium was switched to Differentiation 3 medium (2% human blood plasma, 3% human serum, 3 U/ml heparin, 10 µg/ml insulin, 200 µg/ml holo-transferrin, 10 ng/ml rhSCF, and 1 U/ml Epo in IMDM) and the cell density was maintained at 100,000 cells/ml. After 4 days, the cells were cultured at a density of 1,000,000 cells/ml in Differentiation 4 medium (2% human blood plasma, 3% human serum, 3 U/ml heparin, 10 µg/ml insulin, 500 µg/ml holo-transferrin, and 0.1 U/ml Epo in IMDM) for an additional 4 days. For culture stage 5, the medium was replaced by Differentiation 5 medium (2% human blood plasma, 3% human serum, 3 U/ml heparin, 10 µg/ml insulin, 500 µg/ml holo-transferrin in IMDM) and the density was increased to 5,000,000 cells/ml. Enucleated RBCs were ready to be used after 3 days.

**Hemoglobin content measurement and histology stain of CD34+ cells.** Cells were stained on slides with May-Grünwald–Giemsa and diaminobenzidine hydrochloride reagents (Sigma-Aldrich GS-500 and D-9015) for morphological analyses. Hemoglobin content was measured at 540 nm wavelength light after incubating one million cells with Drabkin's reagent (RICCA Chemical Company, 2660-16). Human hemoglobin (Sigma-Aldrich H7379) was used for the standard curve to calculate hemoglobin amounts from the O.D. 450 nm value.

**Antibodies for western blotting to detect human CD34+ cell proteins.** Antibodies used were anti-XK(Thermo Fisher Scientific, PIPA540782), anti-EPB41 (Abcam, ab54597), anti-MPP1 (Abcam, ab96255), anti-CD47 (Abcam, ab3283), anti-band 3 (Santa Cruz Biotechnology, sc-133190), anti-Integrin β1 (Santa Cruz Biotechnology, sc-18887), anti-Glycophorin C (Santa Cruz Biotechnology, sc-59183), anti-GPA (Santa Cruz Biotechnology, sc-59182), and anti-Kell (Santa Cruz Biotechnology, sc-271070). All antibodies were used at 1:1,000 dilution. Original images of western blottings presented in the manuscript are shown in Supplementary Fig. 5.

**Calculation of the number of recombinant proteins on murine RBCs.** Western blottings were performed with anti-myc-Tag (9B11) mouse monoclonal antibody (Cell Signaling no. 2040, 1:1,000 dilution). In all, 10, 50, and 100 ng myc proteins were used as a quantification standard. To calculate the number of recombinant proteins per cell, a linear plot was generated from the reference myc signal intensities and the number of chimeric RBC proteins expressed per RBC was then derived from the plot. As determined by the myc signal intensity in the western blotting with an anti-myc antibody and as quantified by ImageJ, 1,000,000 murine GPA-VNA/A cells contained 500 ng GPA-VNA/A proteins. The molecular weight of GPA-VNA/A is 60,000. Using Avogadro's number, calculation shows that each red cell expresses 4,600,000 GPA-VNA/A proteins. Similarly, 1,000,000 Kell-VNA/A cells contained 400 ng Kell-VNA/A proteins, of molecular weight 110 kD. Again using Avogadro's number, calculation shows that each red cell expresses 2,190,000 Kell-VNA/A proteins.

**Isolation of erythroid progenitors from murine fetal liver cells.** Enriched erythroid progenitors were purified from E14.5 C57BL/6J mouse embryos and cultured in vitro for erythroid differentiation following a protocol described in detail previously[32]. Briefly, pregnant C57BL/6J mice at embryonic day 14.5 were killed by $CO_2$ asphyxiation and the embryos were collected. The fetal livers were isolated and suspended in PBS with 2% FBS and 100 µM EDTA. Mature RBCs in the cell suspension were lysed by incubation for 10 min with an ammonium chloride solution (Stemcell). Following the manufacturer's protocol, lineage-negative cells were obtained after magnetic depletion of lineage-positive cells using the BD Pharmingen Biotin MouseLineage Panel (559971; BD Biosciences) and BD Streptavidin Particles Plus-DM (557812; BD Biosciences). These lineage-negative fetal liver cells were enriched >90% for erythroid progenitors.

**Viral infection and culture of murine erythroid progenitors.** MSCV-based retroviruses were produced and used to infect erythroid progenitors following a previously described protocol[44]. Briefly, after isolation, lineage-negative fetal liver cells were plated in 24-well plates at 100,000 cells per well, covered by 1 ml virus containing supernatant, and centrifuged at 500 g for 90 min at 30 °C. After this spin-infection, the virus supernatant was replaced with erythroid maintenance medium (StemSpan-SFEM; StemCell Technologies) supplemented with 100 ng/ml recombinant mouse SCF (R&D Systems), 40 ng/ml recombinant mouse IGF1 (R&D Systems), 100 nM dexamethasone (Sigma), and 2 U/ml erythropoietin (Amgen) and cultured at 37 °C. GFP+ cells were sorted by flow cytometry after 16 h and cultured for another 48 h in erythroid differentiation medium (IMDM containing 15% (vol/vol) FBS (Stemcell), 1% detoxified bovine serum albumin (BSA; Stemcell), 500 µg/ml holo-transferrin (Sigma-Aldrich), 0.5 U/ml Epoetin (Epo; Amgen), 10 µg/ml recombinant human insulin (Sigma-Aldrich), and 2 mM L-glutamine (Invitrogen)) at 37 °C.

**Transplantation of mouse fetal liver cells.** C57BL/6J (Jackson Laboratory) or CD-1 (Charles River) mice were subjected to lethal irradiation of 1050 rad carried out in a Gammacell 40 irradiator chamber (Nordion International Inc.) 1 day before transplantation. A total of 1,500,000 virally infected murine erythroid progenitors from mouse fetal liver cultured in erythroid maintenance medium, produced as described in the previous paragraph, were harvested and resuspended in 100 µl PBS[32]. These cells were then injected retro-orbitally into an irradiated mouse. Mice were bled for examining chimeric protein expression and further analysis 5 weeks after transplantation. Mice were bled at the indicated time points for performing CBC analysis on a SIEMENS ADVIA 2120i machine.

**Transfusion and in vivo survival of GPA-VNA RBCs.** A total of 200 µl blood from transplanted mice containing RBCs expressing the GPA-VNA chimera was collected into heparinized tubes (Fisher Scientific, 365965). RBCs were washed twice in PBS and resuspended in PBS. Labeling was then carried out with 5 µM CellTrace violet dye for 20 min at room temperature (Life Technologies). FBS (10%) in PBS was then added to RBCs for quenching the staining reaction. Violet-labeled RBCs were washed twice with PBS and resuspended in 200 µl sterile PBS for intravenous injection into recipient mice. An equal volume of unmodified RBCs, similarly stained, served as a control.

A drop of blood, ~20 µl, was collected into heparinized tubes by retro-orbital bleeding 1 h after transfusion at day 0 and every 3–4 days for 1 month as indicated in the text. These blood samples were washed once with FACS buffer and then subjected to staining with anti-Ter119-APC and anti-myc tag-PE for 30 min on ice. Samples were washed twice with FACS buffer prior to analyses on a FACS Fortessa flow cytometer for violet fluorescence and for GFP, Ter119, and myc signals.

**ciBoNT/A flow cytometry and ELISA assays for toxin binding to RBCs and quantifying toxin in serum.** In all, 200 µl blood from transplanted mice that contain RBCs expressing the GPA-VNA chimera was collected into heparinized tubes and stained with CellTrace violet dye as described above (Figs 3a–c). After resuspending the violet-stained RBCs in 200 µl PBS, RBCs were further incubated with 1 µg catalytically inactive recombinant BoNT/A (ciBoNT/A) on ice for 40 min prior to intravenous injection into recipient mice. An equal volume of unmodified RBCs, also stained and incubated with ciBoNT/A, served as a control. One drop of

blood was collected into heparinized tubes by retro-orbital bleeding 1 h after transfusion (termed day 0) and every 3–4 days for 14 days as indicated in the text and figure legend. Samples were washed once with FACS buffer and then subjected to staining with anti-myc tag-PE to detect surface-expressed VNAs and anti-S-tag-APC (VWR, 10065-802) to indirectly detect ciBoNTA by incubation with the bispecific BoNT/A-binding heterodimer, ciA-F12/D12[3], which carries the S-tag. Samples were washed twice with FACS buffer prior to analyses on a FACS Fortessa flow cytometer for violet fluorescence, GFP, and S-tag (indicating toxin that is bound to the RBC surface) signals.

In a parallel experiment to measure ciBoNT/A in serum of transfused mice (Fig. 3a), 200 µl blood from transplanted mice that contain RBCs expressing the GPA-VNA chimera was also collected into heparinized tubes. RBCs were washed twice in PBS and resuspended in PBS to a volume of 200 µl. As above, these RBCs were incubated with 2 ng ciBoNT/A on ice for 40 min prior to intravenous injection into recipient mice. The recipient mice were then bled 1 h after transfusion (day 0) and every week for 1 month as indicated in the text. In all, 100 µl blood was collected into a serum separator tube (Fisher Scientific, 02675185), and blood samples were allowed to clot at room temperature for 15–30 min. Serum and clots were then separated by centrifugation at 10,000 g for 10 min at 4 °C.

ELISA plates coated with polyclonal anti-BoNT/A (Metabiologics, Inc.) were blocked with blocking buffer (0.05% Tween20 + 2% BSA in PBS) for 2 h at room temperature prior to addition of 10 µl serum samples in 90 µl PBS; the incubation period was 3.5 h at room temperature, followed by four washes with PBS. Plates were then incubated with VHH ciA-D12[3] against BoNT/A, which carries an E-tag, and anti-E tag-horseradish peroxidase (HRP; Bethyl Laboratories, A190-133P), both at 1:1,000 dilutions in blocking buffer for 1 h. Plates were then washed again four times with PBS and developed with 3,3′,5,5′-Tetramethylbenzidine (TMB) liquid substrate reagent (Sigma). Reactions were stopped with 1 N HCl and read at 450-nm absorbance. As a quantification standard, we used serially diluted 1 µg catalytically inactive recombinant BoNT/A.

**ELISA for detecting antibody responses against VNA/A.** Freshly bled 200 µl GPA-VNA/A RBCs and WT C57BL/6J RBCs were washed once with DMEM medium, twice in PBS, and resuspended in sterile PBS to make 200 µl final volume for intravenous injection into C57BL/6J recipient mice. We also transfused a separate cohort of C57BL/6J recipient mice with equimolar amounts of recombinant VNA/A in 200 µl PBS. Three and 4 weeks later, second and third transfusions were carried out in a similar fashion, using the same cohort of recipient mice. Serum samples were collected from these mice 5 days after the last transfusion. Ninety-six-well plates were coated with 10 µg/ml recombinant VNA/A in PBS overnight at 4 °C and blocked in blocking buffer (10% heat inactivated FBS in PBS) prior to addition of 5 µl serum samples in 180 µl blocking buffer. Incubation with test serum was for 3 h at room temperature. Plates were washed four times with PBS, incubated with goat anti-mouse immunoglobulin G–HRP (Southern Biotech) at 1:10,000 in blocking buffer for 1 h, and developed with TMB liquid substrate reagent (Sigma). The reaction was stopped with 1 N HCl and absorbance was read at 450 nm.

**Neuron culture and BoNT/A neutralization assay.** We followed a previously described protocol[3]. In brief, rat (Sprague Dawley strain) cortical neurons were prepared from E18 to E19 embryos. Dissected cortices were dissociated with papain following the manufacturer's instructions (Worthington Biochemical, Lakewood, NJ, USA), and cells were plated on poly-D-lysine coated 24-well plates at 250,000 cells per well in 1 ml Neurobasal medium (Thermo Fisher, Cat. No. 21103-049) supplemented with 2% B27 (Thermo Fisher, Cat. No. 17504-001) and Glutamax (Thermo Fisher, Cat. No. 35050-061). Experiments were carried out using DIV (days in vitro) 11 neurons.

RBCs were pelleted (500 g, 5 min) and resuspended in conditioned neuron culture medium taken from neuron culture plates (0.5 ml per well), and 20 pM BoNT/A was added. The mixture of RBCs and BoNT/A was incubated at 37 °C for 30 min and then added to the original well containing the cultured neurons. Neurons were cultured further for 9 h. Neuron lysates were then collected in 100 µl RIPA buffer (50 mM Tris, 1% NP40, 150 mM NaCl, 0.1% sodium dodecyl sulfate (SDS), plus a protease inhibitor cocktail (Millipore)). Lysates were centrifuged for 10 min at 20,000 g at 4 °C. Supernatants were subjected to SDS–polyacrylamide gel electrophoresis and immunoblot analysis using an SNAP25 antibody[46] and the enhanced chemiluminescence method[47].

**Mouse survival assay.** Our mouse studies were conducted under animal protocols approved by the Division of Comparative Medicine at MIT or the Tufts University IACUC. Eight-week-old female CD-1, C57BL/6J, or NOD/SCID (Jackson Laboratory) mice were weighed before toxin injection and blood transfusion. The average weight of the mice was used to calculate the lethal dose of BoNT/A toxin (Metabiologics, Inc.); the $LD_{50}$ of BoNT/A we employed for C57BL/6J was 1.5 pg/g, which was measured under our animal protocol. The $LD_{50}$ of BoNT/A for CD-1 mice is 1.86 pg/g. The $LD_{50}$ of BoNT/B is 1.58 pg/g. When co-administered with toxin, wild-type or engineered RBCs were transfused in a 200 µl PBS solution retro-orbitally (C57BL/6J mice) or intravenously (CD-1 mice) 30 min–2 h prior to

administration of BoNT/A. Mice were observed 2–6 times/day for 7 days. Mice with symptoms such as difficulty moving, open-mouth breathing, and wasp like narrow waist were killed and scored as moribund.

**Transfusion and in vivo survival of in vitro-differentiated human reticulocytes.** At day 23, in vitro-differentiated human reticulocytes were collected, counted, washed once in PBS, and resuspended in PBS. Labeling of these human reticulocytes was carried out using 5 µM carboxyfluorescein succinimidyl ester (CFSE) according to the manufacturer's instructions (Life Technologies). FBS (10%) in PBS was then added to the reticulocytes for quenching the staining reaction. CFSE-labeled reticulocytes were then washed twice with PBS and resuspended in 200 µl sterile PBS for intravenous injection into NOD/SCID mice at 250,000,000 reticulocytes per mouse. These recipient mice had been intraperitoneally injected with 100 and 50 µl clodronate liposomes at 3 days and 1 day prior to transfusion, respectively. A drop of blood, ~20 µl, was collected into heparinized tubes by retro-orbital bleeding for 1 week starting at 1 h after transfusion at day 0 and at five more time points as indicated in the text. These blood samples were washed once with FACS buffer and then subjected to staining with anti-CD235A-APC for 30 min on ice. Samples were washed twice with FACS buffer prior to analyses on a FACS Fortessa flow cytometer for CFSE and CD235A signals.

**Data availability.** The data that support the findings of this study are available from the corresponding author upon reasonable request

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

## Acknowledgements

This work was sponsored by the Defense Advanced Research Projects Agency contract no. HR0011-12-2-0015 and the prior VHH and VNA development was sponsored by U54 AI057159 (CBS). N.P. was sponsored by grants from the Schlumberger Foundation Faculty for the Future and the Howard Hughes Medical Institute International Student Research Fellowship. The content of the information presented here does not necessarily reflect the position or the policy of the Government, and no official endorsement should be inferred. We thank members of Lodish laboratory, especially Jiahai Shi, for fruitful discussions, and Tony Chavarria and Ferenc Reinhardt for mouse husbandry. We also thank Michelle Debatis for her technical support in the Shoemaker laboratory. We gratefully acknowledge Dr. Konstantin Ichtchenko and his laboratory for their contribution of the ciBoNTA used in the pharmacokinetic studies. We thank Tom DiCesare for help with illustrations.

## Author contributions

N.-J.H. and N.P. designed the experiments and analyzed the data. N.-J.H., N.P., C.B.S. and H.F.L. wrote the manuscript. N.-J.H. and N.P. performed most of the experiments with help from J.M. for CD1 mice survival assay, S.Z. and C.B.S. for in vitro neuron assay and R.D. and V.S. for preparing experimental materials.
