## [Peer Review File · Nature Communications]

Reviewers' comments:

Reviewer #1 (expert in RBC biotechnology; Remarks to the Author):

The manuscript describes the generation of mouse RBCs in vivo and in vitro expressing chimeric GPA and Kell with VHH against BoNT/A (and B), and then goes on to create human RBCs in vitro also expressing chimeric GPA with VHH against BoNT/A. They demonstrate that mice generating a small number of these engineered RBCs in vivo, or transfused with engineered mouse or human RBCs can survive challenge with BoNT/A.

The concept, whilst not novel, is definitely interesting, and uses a recombinant approach, engineering the cells to express VHH as part of a RBC membrane protein rather than attaching to a RBC membrane protein, to improve protection to infection

The manuscript is reasonably well written, but there a quite a few inaccuracies and omissions of required information. However the work itself is substantially incomplete and requires substantially more data as described below, as well as explanations/further explanations of many of the results.

Introduction

This should include description of similar studies in the literature which also address creating model systems to protect against BoNT infection using alternative approaches, but with at least similar outcomes. For example the study by Adekar 2011 describes a novel approach to link BoNT-specific antibodies to GPA on the RBC membrane which gives mice protection from a lethal BoNT/A1 dose, and demonstrates toxin cleared from serum. This study is not acknowledged by the authors, even though they are using the same concept, and as such comparison and the advantages of the different approaches should be addressed. In addition, the study by Mukherjee 2014 (with some shared authors to the present study) achieved 5 days extended serum life of VHA to BoNTA/1 by binding an albumin-binding-peptide to an anti-BoNT/A1 VNA with mice surviving challenge to toxin 5 days post administration, and at least a similar protection duration to the present study was achieved by using adenoviral transduction of a VHH to BoNT/A.

Results

The authors need to demonstrate far more conclusively that the modification to GPA and Kell do not effect the RBCs. Just showing that they still enucleate with a culture system that is clearly not optimal as enucleation rates are only around 30% even for control cells is not adequate. Data comparing proliferation rates, differentiation rates (with cell morphology) should be included and importantly the authors need to show that the GPA and Kell containing red blood cell membrane complexes are not perturbed by inclusion of the chimeric proteins, and that the expression level and time of expression during erythropoiesis of chimeric GPA and Kell vs normal GPA and Kell, as well as other membrane proteins are not effected. Importantly is the deformability of the chimeric protein containing retics altered? The authors cannot claim that their cells underwent 'normal' erythropoiesis without looking at least these parameters.

How was the copy number of chimeric GPA and Kell in the membrane of the mouse cells calculated?

Page 7 line 10 I assume should say Kell-VNA/A?

Do the engineered RBCs get retained in the spleen or other organs in the in vivo or transfused mice?

Importantly, are antibodies raised to the chimeric proteins in vivo.

Are the anti-BoNT VHHs generated specific to BoNT, or do they cross react with other proteins?

The statement page 7 line 18 is inaccurate as only around 60% of mice survived BoNT/B challenge.

Transfusion with red blood cells from chimeric mice:

How many GPA-VNA/A+ve cells were transfused? What was the duration of the effect? Why was the effect far less protective than that found for the chimeric mice, even though there were more chimeric cells in circulation after transfusion than found in the chimeric mice? This, and the

viability of the approach for protection in humans needs to be taken into consideration and discussed.

By day 28 there are a tiny fraction of the transfused GPA-VNA/A+ve remaining in circulation, however the mice are still protected from challenge with 10LD50 BoNT/A. If so few cells can protect against this challenge, it needs to be explained why the significantly greater number of cells present at 24hrs are unable to protect against the 10x greater challenge of 100 LD50 BoNT/A – the number of cells present at 24hrs is clearly more than 10x that present at 28 days.

Fate of BoNT/A and engineered cells in mice:

Why was a true time zero reading not performed showing proportion of bound ciBoNT/A before transfusion? Surely most of the binding occurred when the cells and toxin were mixed, not as the authors state after 1hr post transfusion.

There are no error bars on the 'time zero' point on the graph in Fig2d.

More time points from transfusion to 1hr post transfusion are required (Fig 2d) as clearly all events occur in this time period.

How do the authors know that 100% of transgenic RBCs had bound ciBoNT/A?

Why is the RBC bound ciBoNT/A cleared more rapidly than the GPA+ cells, even though the authors have stated that 100% of transgenic RBCs have bound ciBoNT/A by 1hr? On the same lines, if as the authors hypothesise, binding of BoNT/A to GPA-VNA/A enhances their degradation by macrophages or dendritic cells why does it take 14 days to clear these cells when they have apparently all bound BoNT/A by time point 0? This hypothesis could have been tested.

Human in vitro generated reticulocytes and in survival in vivo in mouse model:

The erythroid culture system used is not improved compared to systems already published (eg see Griffiths et al 2012)

The authors do not demonstrate that differentiation was synchronous – flow cytometry at different time points in culture should be performed for a range of RBC membrane proteins, and in particular morphological analysis to determine synchronicity of the culture.

How were the reticulocytes isolated?

A reference for the 'normal' size of endogenous human reticulocytes must be included for the authors to categorically state the size of their in vitro generated reticulocytes is normal.

How was the hemoglobin content of the cells measured, it can absolutely not be done just by staining the cells as claimed in the manuscript.

What proportion of the injected cells survived 10mins, 1hr, 1 day etc in the mice? How does this compare with injection of normal endogenous human reticulocytes or even donor RBCs? This data is essential to understand if the in vitro generated cells behave normally.

Why do the human cells only survive 7 days in the mice?

Engineered human RBCs:

How was the number of copies of hGPA-VNA/A calculated?

Mapping expression of just GPA does not demonstrate that the cells differentiated normally, and clearly there was a reduction in enucleation rate. How did the proliferation rate, differentiation rate compare with non-engineered cells? Morphology of the cells throughout erythropoiesis needs to be shown.

As for the mouse cells a range of parameters need to be examined to determine if the engineered RBCs are normal eg. The authors need to show that the GPA containing red blood cell membrane complexes are not perturbed by inclusion of the chimeric proteins, and that the expression level and time of expression during erythropoiesis of chimeric GPA vs normal GPA, as well as other membrane proteins are not effected. Importantly is the deformability of the chimeric protein containing retics altered?

What was the survival rate of engineered human reticulocytes in the mice compared to non engineered at 10mins, 1hr, 1 day etc?

Do the engineered cells get retained in the spleen or other organs?

Assuming numbers are the same as for the non engineered cell, clearly the tiny number of cells remaining at day 7 are still completely adequate to protect against a 10LD50 dose of BoNT/A. Is this

surprising to the authors? Importantly, how long did this protection last? Presumably if such a tiny number of cells can still protect at day 7 post transfusion, the mice at day 1 should be able to survive infection with a higher dose? Why was this not performed?

Reviewer #2 (expert in nanobodies; Remarks to the Author):

The authors make genetic fusion of VHH (with specificity for botulinum neurotoxin serotype A (BoNT/A) to glycophorin or Kell RBC membrane proteins. These chimeric gene constructs in retroviral vectors are introduced in embryonic mouse fetal liver red cell progenitors. The RBC exposing the chimeric VHH constructs protect neuronal cells from BoNT/A toxicity.

Mice receiving a limited RBC transfusion also seem to be protected against high doses of the BoNT/A. This protection holds for very long times (1 month), which is surprising as autonomous VHH are rapidly cleared from blood ($t_{1/2}$ within 30 min). Obviously, several methods have been applied to increase the circulation half-life time of VHH in blood (by increasing the hydrodynamic volume of VHH or by attaching the VHH to another VHH that associates with an abundant blood protein such as serum albumin or IgG). However, the strategy explored in this study surpasses by far the other methods. The strategy to choose for retroviral vectors and RBC as transgenic host is also clever as mature erythrocytes have lost their nucleus during erythropoiesis. The method was also tested on human CD34+ stem cells.

Experiments are well described (in M&M section) and are performed scientifically correct and data are clearly presented (except one minor comment for Figure 3b where the % of cleaved SNAP25 is shifted and not properly aligned under the wells). With the information given, many scientists in various laboratories will be able to repeat and amend this work and/or introduce further developments. The work is considered to be novel and innovative, and is expected to be a catalysator for further studies.

Previous work is correctly cited.

Overall this work is interesting for researchers with a broad interest in infections, intoxications and innovative therapeutic tools.

Reviewer #3 (expert in RBC engineering; Remarks to the Author):

Major claims of the paper:

Engineered RBCs can be efficient carrier for new therapeutics increasing blood life-span of the same, efficient protection of the host when the therapeutic agent is directed against bacterial toxins, and possibly translatable for other therapeutic applications.

Are they novel and of general interest?

Dr. Harvey Lodish have already published (PNAS 2014,111:10131-10136) in vitro engineered erythroid precursors to express two of the most abundant RBC membrane proteins (Glycophorin A, GPA, and the blood group antigen Kell) with respectively an extracellular N terminus (GPA) or C-terminus (Kell). These erythroid precursors could then be differentiated in vitro to mature RBCs and the extracellular terminus labeled in a sortase-catalyzed reaction with different payloads, including a single domain antibody. Dr. Charles Shomaker have extensively published on single domain camelid antibodies as therapeutic agents for the protection against a number of different toxins. However, the present publication recognizes some of the limits of the previous approaches and provide new solutions to the problem of improving efficacy of a combined treatment that take

advantage from the possibility of engineering RBCs precursors using single domain camelid antibodies fused to the most abundant RBC membrane proteins for long term protection. The approach could be of general interest also for others in the field. The data presented represent a "proof-of-concept" and must be appreciated, however, additional points should be considered and discussed to fully justify the conclusions.

Major issues

The modification of RBC by engineering membrane proteins with different additional foreign sequences (i.e. the single domain antibody) once transfused in a compatible host should induce alloimmunization. RBC alloimmunization could be a serious complication especially in case of repeated administrations. When alloantibodies are formed, in many cases, RBCs expressing the antigen in question can no longer be safely transfused. The Authors have not considered this potential safety issue and report only in vivo experiments that consist of a single RBC administration, frequently in irradiated mice or in macrophage-depleted NOD/SCID mice, without evidence for the safety and efficacy of repeated administrations. It should be useful to show that repeated administrations of the selected candidates are safe and that does not induce humoral or cellular responses against the engineered sequences. As an alternative, the possible induction of antibodies against the engineered RBCs (three-four weeks after engineered RBC administration) should be measured.

Minor points:

a) The Authors (Introduction, lines 3 and 4 from bottom) suggest that attachment of large number of cargoes on the membrane of RBCs is not provoking adverse immune reactions. This is not completely true and usually depends on the density of the cargoes. As an example (Transfusion. 2011 May;51(5):1047-57; Anal Biochem. 1996 Oct 1;241(1):109-19) it has been reported that biotin density on RBC can affect RBC survival in circulation (biotin is usually considered very safe!), or induce both humoral and cellular responses that are higher than administration of the soluble antigen (Vaccine. 2003 May 16;21(17-18):2073-81). The sentence should be modified highlighting also the possible risks.

b) Pag.6 lines 6 and 7 from bottom: the estimate number of about 5,000,000 copies of chimeric proteins on RBC surface is stable upon final differentiation? (i.e. after reticulocyte maturation)

c) Pag.7 lines 9 and 10 from top: the reported percentage of myc positive RBC determined after six weeks from bone marrow reconstruction (2.96%; 14.26%) remain constant at the time of repeated challenge with increasing doses of BoNT/A or increases? In other words, the percentage of RBC expressing GPA-VNA/A in circulation could increase over time and contribute to explain the higher resistance of multiple challenged mice to BoNT/A? Please provide percentage of myc expressing cells in circulation at each challenging time and possibly provide hematological data documenting bone marrow reconstitution over time.

d) Pag.8 line 8-9 from top: it should be mentioned that instead, fully human or humanized anti botulinum antibodies, showed in vivo a t_{1/2} ranging from 2.5 to 26.9 days depending on dose and antibody (Antimicrob. Agents Chemother. September 2014 58:5047-5053)

e) Pag.10 line 7 from bottom: survival in circulation for 7 days (estimate t_{1/2} about 2.5 days) in macrophage-depleted NOD/SCID mice seems to be very low. What is the t_{1/2} of unprocessed (native) human RBC under the same conditions? Please provide comparison data to discriminate the role of the host vs the role of the in vitro differentiation of transduced erythroblasts.

f) Pag.11 line 1 from top: please comment on the reduced expression on human RBC vs murine RBC of the constructs.

g) DISCUSSION: the discussion section is very speculative and several conclusions are not fully supported by data. Two key issues should be mentioned in the discussion at least, the safety issue including possible alloimmunization (I have already commented above) and the transfer in the recipient of retroviral transduced cells. While the Authors have documented the expansion and maturation in vitro to be quite efficient, still a significant number of nucleated cells are present (about 30% of engineered human RBC, pag.11 line three from top).

h) Pag.13 line 5-6: biological production of recombinant protein is certainly not an issue nowadays!

i) Pag.13 line 7 from bottom: clearance of BoNT/A bound to RBC is a relevant issue and should be evaluated in more details since the antibody is not neutralizing the toxin and the accumulation of

the same in some compartments could represent an important issue.

j) Pag.14 speculations about the use of CR1 as a coupling site for VHH, the use of more than three VHH domains on the same cell or the expression of functional cargos inside the RBC are speculations that should be contained in few lines or less.

k) Pag.15, last line from bottom: the estimate of 26 days survival of cRBCs should be taken carefully since it represent only one donor, the labelled cells were retics and the Cr51 elution was not estimated but extrapolate from other studies. As a consequence, the last sentence in the Discussion should mention the need of additional data to confirm the expectations!

Reviewers' comments:

Reviewer #1 (expert in RBC biotechnology; Remarks to the Author):

The manuscript describes the generation of mouse RBCs in vivo and in vitro expressing chimeric GPA and Kell with VHH against BoNT/A (and B), and then goes on to create human RBCs in vitro also expressing chimeric GPA with VHH against BoNT/A. They demonstrate that mice generating a small number of these engineered RBCs in vivo, or transfused with engineered mouse or human RBCs can survive challenge with BoNT/A.

The concept, whilst not novel, is definitely interesting, and uses a recombinant approach, engineering the cells to express VHH as part of a RBC membrane protein rather than attaching to a RBC membrane protein, to improve protection to infection

The manuscript is reasonably well written, but there a quite a few inaccuracies and omissions of required information. However the work itself is substantially incomplete and requires substantially more data as described below, as well as explanations/further explanations of many of the results.

Introduction

This should include description of similar studies in the literature which also address creating model systems to protect against BoNT infection using alternative approaches, but with at least similar outcomes. For example the study by Adekar 2011 describes a novel approach to link BoNT-specific antibodies to GPA on the RBC membrane which gives mice protection from a lethal BoNT/A1 dose, and demonstrates toxin cleared from serum. This study is not acknowledged by the authors, even though they are using the same concept, and as such comparison and the advantages of the different approaches should be addressed. In addition, the study by Mukherjee 2014 (with some shared authors to the present study) achieved 5 days extended serum life of VHA to BoNTA/1 by binding an albumin-binding-peptide to an anti-BoNT/A1 VNA with mice surviving challenge to toxin 5 days post administration, and at least a similar protection duration to the present study was achieved by using adenoviral transduction of a VHH to BoNT/A.

Thank you for pointing out the lack of these two important references. We have included these references in the second paragraph of the manuscript.

Results

1. The authors need to demonstrate far more conclusively that the modification to GPA and Kell do not effect the RBCs. Just showing that they still enucleate with a culture system that is clearly not optimal as enucleation rates are only around 30% even for control cells is not adequate. Data comparing proliferation rates, differentiation rates (with cell morphology) should be included and importantly the authors need to show that the GPA and Kell containing red blood cell membrane complexes are not perturbed by inclusion of the chimeric proteins, and that the expression level and time of expression during erythropoiesis of chimeric GPA and Kell vs normal GPA and Kell, as well as other membrane proteins are not effected. Importantly is the deformability of the chimeric protein containing retics altered? The authors cannot claim that their cells underwent 'normal' erythropoiesis without looking at least these parameters.

The 30% enucleation is from in vitro cultured mouse fetal liver cells. We understand that enucleation is not very high and that this number is similar to other reports. (*J. Vis. Exp.* (91), e51894, doi:10.3791/51894 (2014).) We did not improve the mouse *in vitro* culture system; however, we include data concerning proliferation, differentiation, and cell morphology in Fig. S1d, e and f. Importantly, the key in vivo results did not make use of mouse red cells made in culture, but rather normal red cells made in transplanted mice (Figure 1 c- e).

The new Figure 5 shows that genetically engineered human RBCs made in culture and expressing GPA-VNA/A or Kell-VNA/A differentiate normally and protect neurons against BoNT/A challenge both in neuronal culture and *in vivo*. Panel A shows a FACS analysis of control and VNA-

expressing human erythroid cells during the culture, demonstrating, as judged by expression of multiple cell surface proteins, that development is unperturbed by chimeric VNA expression. Panel B shows, by both FACS analysis and histological staining, that enucleation is slightly reduced but that cell morphology is unperturbed by GPA-VNA/A expression. And Panel C shows that cell proliferation in our culture system is unperturbed by GPA-VNA/A expression. Figure S4b shows Western blots of several key erythrocyte proteins, demonstrating that expression of these proteins is normal in our human red cells produced in culture. And Figure S4c shows that human red cells made in culture, expressing or not GPA-VNA/A, survive in macrophage- depleted NOD/SCID mice as long as do normal human red cells. Collectively our data shows that human GPA-VNA/A cells are similar to control cells, indicating that our cell engineering caused negligible changes to erythropoiesis; however, we admit that lentiviral delivery is a perturbation to the culture system. We also understand that the reticulocytes we produced in culture are not mature red cells, but they can survive in vivo for at least 7days. And these cells injected in NOD/SCID mice should mature to normal red cells according to the work reported in *Haematologica* 102, 476-483 (2017).

2. How was the copy number of chimeric GPA and Kell in the membrane of the mouse cells calculated?

Please see the new text we added to the legend for Figure S1b

3. Page 7 line 10 I assume should say Kell-VNA/A?

We are sorry for the mistake. It has been changed accordingly.

4. Do the engineered RBCs get retained in the spleen or other organs in the in vivo or transfused mice?

We know that the engineered RBCs have an equivalent half-life to unmodified RBCs (Fig. 2C); therefore retention in spleen or other organs can be safely assumed to be similar to those of unmodified RBCs.

5. Importantly, are antibodies raised to the chimeric proteins in vivo.

Please see the new Fig. 2h. Antibodies raised against the chimeric proteins expressed on RBCs are at least two orders of magnitude fewer than the antibodies raised against the same amount of recombinant VNA-A.

6. Are the anti-BoNT VHHs generated specific to BoNT, or do they cross react with other proteins?

Those generated specifically to BoNT/A show no evidence of cross-reactivity based on lack of binding to closely related BoNTs, such as BoNT/B. (Please see *PLoS One* 7, e29941 (2012) for details). Hence, we did not perform broad cross-reactivity studies with other antigens.

7. *The statement page 7 line 18 is inaccurate as only around 60% of mice survived BoNT/B challenge.*

Thank you for the correction. We added “60%” in the statement.

8. *Transfusion with red blood cells from chimeric mice:
How many GPA-VNA/A+ve cells were transfused?*

We injected 100 µl whole blood containing 6.22±0.25% of RBCs displaying surface GPA-VNA/A. There are 10⁷ cells/µl of whole blood. Therefore, as now noted in the text, we injected 6.22*10⁷ GPA-VNA/A expressing RBCs.

9. *What was the duration of the effect? Why was the effect far less protective than that found for the chimeric mice, even though there were more chimeric cells in circulation after transfusion than found in the chimeric mice? This, and the viability of the approach for protection in humans needs to be taken into consideration and discussed.*

In the case of transfusion, we usually injected only up to 200 µl (except for Fig. 2b) of whole blood into a mouse, which has a total of 2.5 mL of blood in its circulation. Therefore, assuming ~6% of the transfused RBCs displaying GPA-VNA/A, in the initial transplanted mouse, this GPA-VNA/A expressing RBCs only make up 0.48% of the mouse' total blood at day 0 of transfusion. This number declines over time since the transfused blood, itself a mixture of old and young RBCs, and, as we show, will be cleared from the circulation with a half-life of about 2 weeks. The fact that resistance to BoNT/A lasted at least 28 days indicates that very small numbers of red cells expressing the GPA-VNA/A suffice to neutralize multiple lethal doses of toxins. Given the longer lifetime of human red cells, our experiments indicate that transfusion of human red cells expressing GPA-VNA/A into humans would induce protection for at least 3 months.

10. *By day 28 there are a tiny fraction of the transfused GPA-VNA/A+ve remaining in circulation, however the mice are still protected from challenge with 10LD50 BoNT/A. If so few cells can protect against this challenge, it needs to be explained why the significantly greater number of cells present at 24hrs are unable to protect against the 10x greater challenge of 100 LD50 BoNT/A – the number of cell present at 24hrs is clearly more than 10x that present at 28 days.*

We reasoned that a bolus amount of toxin might escape neutralization before the VNA/A RBCs have the chance to neutralize all of the toxin. We therefore injected 400 ul of GPA-VNA/A blood and challenged the mice after 1hr. The mice were protected, as the new Fig. 2b shows. Also, we observed that mice undergoing a 1000 LD₅₀ challenge have more severe symptoms (total lack of movement and wasp waist) than mice receiving 100LD₅₀ (their leg muscles are weaker than untreated mice.) The mice under 100 LD₅₀ challenge were judged as dead according to our animal protocol since we

needed to euthanize them if they were not fully protected. Our results suggest that the amount of RBCs we used is nearly enough to protect mice from 100 LD₅₀.

11. Fate of BoNT/A and engineered cells in mice:

Why was a true time zero reading not performed showing proportion of bound ciBoNT/A before transfusion? Surely most of the binding occurred when the cells and toxin were mixed, not as the authors state after 1hr post transfusion.

There are no error bars on the 'time zero' point on the graph in Fig2d.

More time points from transfusion to 1hr post transfusion are required (Fig 2d) as clearly all events occur in this time period.

Performing a zero time control after a transfusion indeed is difficult! Both the error bar of control blood and the GPA-VNA/A for "time 0" in Fig. 2d were shown; please see the black bars of controls at 1.2 and 1.5 and the error bars of GPA-VNA/A overlapping with the red square. In the new Figure S3 we studied much earlier time points, close to the real time zero. Panel A shows a loss of only ~20% of the S-tag, detecting RBC- bound ciBoNT/A, during the first hour following transfusion of GPA-VNA/A expressing RBCs red cells. And Panel B shows no loss of the GPA-VNA/A expressing RBCs themselves during the first hour following transfusion of GPA-VNA/A expressing RBCs red cells. Thus the use of the 1 hour time point in the experiment in Figure 2d is appropriate.

12. How do the authors know that 100% of transgenic RBCs had bound ciBoNT/A?

Due to the limitation of the detection method, i.e. flow cytometry resolution, we do not know, nor did we claim, that 100% of transgenic RBCs had bound ciBoNT/A. We tried to oversaturate the incubation process to drive the binding to completion and use comparison with normal RBCs as a negative control.

13. Why is the RBC bound ciBoNT/A cleared more rapidly than the GPA+ cells, even though the authors have stated that 100% of transgenic RBCs have bound ciBoNT/A by 1hr? On the same lines, if as the authors hypothesis, binding of BoNT/A to GPA-VNA/A enhances their degradation by macrophages or dendritic cells why does it take 14 days to clear these cells when they have apparently all bound BoNT/A by time point 0? This hypothesis could have been tested.

Unfortunately, we have no answer for this phenomenon, in large measure because we do not understand precisely how red cells are designated for degradation. Nor do we know the precise identity of the phagocytic cells that normally degrade aged red cells. This is indeed an area of interest our future studies, in which we are developing biosynthetic labeling of our engineered RBCs and/or ciBoNT/A that is sensitive enough to trace the fate of red cells *in vivo*.

14. *Human in vitro generated reticulocytes and in survival in vivo in mouse model: The erythroid culture system used is not improved compared to systems already published (eg see Griffiths et al 2012)*

It is our fault for not including papers from this group in our manuscript. We compared the system described in that paper (Blood 119, 6296-6306, 2012) with our system. We found that our enucleation efficiency is routinely >90% and whereas the earlier published system shows 60~85% enucleation. We acknowledge that our system is similar in terms of proliferation compared with the system described in Griffiths et al 2012 and we have now referenced this paper in this manuscript.

15. *The authors do not demonstrate that differentiation was synchronous – flow cytometry at different time points in culture should be performed for a range of RBC membrane proteins, and in particular morphological analysis to determine synchronicity of the culture.*

The new Figure 5 shows that genetically engineered human RBCs made in culture and expressing GPA-VNA/A or Kell-VNA/A differentiate normally and protect neurons against BoNT/A challenge both in neuronal culture and *in vivo*. Please see Fig. 5a showing flow cytometry data at different time points and demonstrating that differentiation indeed is synchronous, and Fig. S4a for protein expression. Morphology data were included in Fig. 4c and Fig. 5b.

16. *How were the reticulocytes isolated?*

We did not isolate reticulocytes.

A reference for the 'normal' size of endogenous human reticulocytes must be included for the authors to categorically state the size of their in vitro generated reticulocytes is normal.

Thank you for pointing out that we were missing a reference. We have included the reference-PLoS One 8, e76062 (2013).

17. *How was the hemoglobin content of the cells measured, it can absolutely not be done just by staining the cells as claimed in the manuscript.*

It was described in Methods in our previous version of this manuscript, in the last paragraph of "Human CD34+ cell culture." In this version we have moved the paragraph to a paragraph "Hemoglobin content measurement and histology stain of CD34+ cells".

18. *What proportion of the injected cells survived 10mins, 1hr, 1 day etc in the mice? How does this compare with injection of normal endogenous human reticulocytes or even donor RBCs? This data is essential to understand if the in vitro generated cells behave normally.*

Please see Fig. 4d; these cells survived for at least 7 days. Also within 1 hour, the survival rate of these cells is similar as unmodified human RBCs (Fig. S4c).

19. Why do the human cells only survive 7 days in the mice?

Sorry for the confusion. It is not that the cells only survived for 7 days. It is the detection limit by FACS. We changed the description to “at least 7 days.” It is well known that human red cells are larger than mouse erythrocytes, and that they are cleared in a few days when transfused into immune competent mice.

Engineered human RBCs:

20. How was the number of copies of hGPA-VNA/A calculated?

Please see the legend to the new Figure S4A

21. Mapping expression of just GPA does not demonstrate that the cells differentiated normally, and clearly there was a reduction in enucleation rate. How did the proliferation rate, differentiation rate compare with non-engineered cells? Morphology of the cells throughout erythropoiesis needs to be shown. The data are included in Fig. 5a, Fig. 5b and Fig. 5C. We believe the reduction of enucleation is due to virus infection instead of GPA-VNA/A genetic modification.

22. As for the mouse cells a range of parameters need to be examined to determine if the engineered RBCs are normal e.g. The authors need to show that the GPA containing red blood cell membrane complexes are not perturbed by inclusion of the chimeric proteins, and that the expression level and time of expression during erythropoiesis of chimeric GPA vs normal GPA, as well as other membrane proteins are not effected. Importantly is the deformability of the chimeric protein containing retics altered?

The new Figure 5 shows that genetically engineered human RBCs made in culture and expressing GPA-VNA/A or Kell-VNA/A differentiate normally and protect neurons against BoNT/A challenge both in neuronal culture and *in vivo*. Specifically, Panel A shows a FACS analysis of control and VNA- expressing human erythroid cells during the culture, demonstrating, as judged by expression of multiple cell surface proteins, that development is unperturbed by chimeric VNA expression.

23. What was the survival rate of engineered human reticulocytes in the mice compared to non engineered at 10mins, 1hr, 1 day etc?

Please see Fig. S4c; the survival rate of these cells is similar as vector only expressing cells and normal human RBCs in 1 hour.

24. Do the engineered cells get retained I the spleen or other organs?

The spleen is the organ where aged/damaged cells are retained. But as noted above, we do not know the precise identity of the phagocytic cells that degrade normal red blood cells. Determining the cells that ingest and degrade engineered red cells is beyond the scope of this paper; much of our current research is now focused on this topic, in part to understand why, as shown in our recent paper (PNAS 2017 114 (12) 3157-3162), many peptides and proteins attached to the surface of red cells induce tolerance to the attached protein rather than an immune response.

25. Assuming numbers are the same as for the non engineered cell, clearly the tiny number of cells remaining at day 7 are still completely adequate to protect against a 10LD50 dose of BoNT/A. Is this surprising to the authors? Importantly, how long did this protection last?

Presumably if such a tiny number of cells can still protect at day 7 post transfusion, the mice at day 1 should be able to survive infection with a higher dose? Why was this not performed?

The purpose in our manuscript is to demonstrate the obvious efficacy of the human GPA-VNA/A RBC. Of course, it would be interesting to know how long protection in the mice can last. We observed that NOD/SCID mice could be protected beyond 1 day post transfusion. However, it is difficult to say more since we do not know if the engineered VNA on human cells are damaged in NOD/SCID mice.

Reviewer #2 (expert in nanobodies; Remarks to the Author):

The authors make genetic fusion of VHH (with specificity for botulinum neurotoxin serotype A (BoNT/A) to glycophorin or Kell RBC membrane proteins. These chimeric gene constructs in retroviral vectors are introduced in embryonic mouse fetal liver red cell progenitors. The RBC exposing the chimeric VHH constructs protect neuronal cells from BoNT/A toxicity.

Mice receiving a limited RBC transfusion also seem to be protected against high doses of the BoNT/A. This protection holds for very long times (1 month), which is surprising as autonomous VHH are rapidly cleared from blood (t_{1/2} within 30 min). Obviously, several methods have been applied to increase the circulation half-life time of VHH in blood (by increasing the hydrodynamic volume of VHH or by attaching the VHH to another VHH that associates with an abundant blood protein such as serum albumin or IgG). However, the strategy explored in this study surpasses by far the other methods. The strategy to choose for retroviral vectors and RBC as transgenic host is also clever as mature erythrocytes have lost their nucleus during erythropoiesis. The method was also tested on human CD34+ stem cells.

Experiments are well described (in M&M section) and are performed scientifically correct and data are clearly presented (except one minor comment for Figure 3b

where the % of cleaved SNAP25 is shifted and not properly aligned under the wells). With the information given, many scientists in various laboratories will be able to repeat and amend this work and/or introduce further developments. The work is considered to be novel and innovative, and is expected to be a catalysator for further studies.

Previous work is correctly cited.

Overall this work is interesting for researchers with a broad interest in infections, intoxications and innovative therapeutic tools.

We are very grateful about these very positive comments from a nanobody expert, as we now think of red cells as “microbodies” that can deliver multiple types of therapeutics into the body for long periods of time!

Reviewer #3 (expert in RBC engineering; Remarks to the Author):

Major claims of the paper:

Engineered RBCs can be efficient carrier for new therapeutics increasing blood life-span of the same, efficient protection of the host when the therapeutic agent is directed against bacterial toxins, and possibly translatable for other therapeutic applications.

Are they novel and of general interest?

Dr. Harvey Lodish have already published (PNAS 2014,111:10131-10136) in vitro engineered erythroid precursors to express two of the most abundant RBC membrane proteins (Glycophorin A, GPA, and the blood group antigen Kell) with respectively an extracellular N terminus (GPA) or C- terminus (Kell). These erythroid precursors could then be differentiated in vitro to mature RBCs and the extracellular terminus labeled in a sortase-catalyzed reaction with different payloads, including a single domain antibody. Dr. Charles Shomaker have extensively published on single domain camelid antibodies as therapeutic agents for the protection against a number of different toxins. However, the present publication recognizes some of the limits of the previous approaches and provide new solutions to the problem of improving efficacy of a combined treatment that take advantage from the possibility of engineering RBCs precursors using single domain camelid antibodies fused to the most abundant RBC membrane proteins for long term protection. The approach could be of general interest also for others in the field. The data presented represent a “proof-of-concept” and must be appreciate, however, additional points should be considered and discussed to fully justify the conclusions.

Major issues

1. The modification of RBC by engineering membrane proteins with different additional foreign sequences (i.e. the single domain antibody) once transfused in

a compatible host should induce alloimmunization. RBC alloimmunization could be a serious complication especially in case of repeated administrations. When alloantibodies are formed, in many cases, RBCs expressing the antigen in question can no longer be safely transfused. The Authors have not considered this potential safety issue and report only in vivo experiments that consist of a single RBC administration, frequently in irradiated mice or in macrophage-depleted NOD/SCID mice, without evidence for the safety and efficacy of repeated administrations. It should be useful to show that repeated administrations of the selected candidates are safe and that does not induce humoral or cellular responses against the engineered sequences. As an alternative, the possible induction of antibodies against the engineered RBCs (three-four weeks after engineered RBC administration) should be measured.

Thank you for raising this essential question. We have performed the suggested experiment in the new Fig. 2h and we have discussed this issue in the fifth paragraph in our discussion section. This new data shows little induction of antibodies to the VNAs expressed on the red blood cells in contrast to strong induction of antibodies against injected VHH protein. Our recent PNAS paper (PNAS 2017 114 (12) 3157-3162) shows that many peptides attached to red cells induce sequence- specific immune tolerance rather than an immune response.

Minor points:

a) The Authors (Introduction, lines 3 and 4 from bottom) suggest that attachment of large number of cargoes on the membrane of RBCs is not provoking adverse immune reactions. This is not completely true and usually depends on the density of the cargoes. As an example (Transfusion. 2011 May;51(5):1047-57; Anal Biochem. 1996 Oct 1;241(1):109-19) it has been reported that biotin density on RBC can affect RBC survival in circulation (biotin is usually considered very safe!), or induce both humoral and cellular responses that are higher than administration of the soluble antigen (Vaccine. 2003 May 16;21(17-18):2073-81). The sentence should be modified highlighting also the possible risks.

We are grateful for this keen insight. As seen in Fig. 2h, GPA-VNA/A RBCs are much less immunogenic compared to injection of the pure VNA/A recombinant protein. This is yet another area of interest in our future studies - to investigate the limit of immune tolerability against RBC cargoes that can lead to divergent immune response to proteins attached to red cells. A more comprehensive study analyzing the effects of cargo number/RBC as well as covalent vs. non-covalent linkage to RBCs is undoubtedly imperative. Nevertheless, we have added several sentences in both Result and Discussion sections to highlight the possible risks.

b) Pag.6 lines 6 and 7 from bottom: the estimate number of about 5,000,000 copies of chimeric proteins on RBC surface is stable upon final differentiation? (i.e. after reticulocyte maturation)

The new Figure S1g shows that 6,000,000 GPA-VNA/A red cells, produced in mice transplanted with VNA/A- expressing progenitors, express ~200 ng Myc-GPA-VNA/A protein, or about 310,000 VNA/A proteins per cell, about 1/15th that found on red cells made in *in vitro* culture.

c) Pag.7 lines 9 and 10 from top: the reported percentage of myc positive RBC determined after six weeks from bone marrow reconstruction (2.96%; 14.26%) remain constant at the time of repeated challenge with increasing doses of BoNT/A or increases? In other words, the percentage of RBC expressing GPA-VNA/A in circulation could increase over time and contribute to explain the higher resistance of multiple challenged mice to BoNT/A? Please provide percentage of myc expressing cells in circulation at each challenging time and possibly provide hematological data documenting bone marrow reconstitution over time.

The average percentage of myc expressing cells in the blood is stable after one month and up to six months post-transplantation, as shown in the new Fig. 1d. We could not detect myc expressing cells in mice that have been challenged with BoNT/A since our biohazard protocol does not allow us to perform analysis of biohazard materials on a flow cytometer.

d) Pag.8 line 8-9 from top: it should be mentioned that instead, fully human or humanized anti botulinum antibodies, showed in vivo a t1/2 ranging from 2.5 to 26.9 days depending on dose and antibody (Antimicrob. Agents Chemother. September 2014 58:5047-5053)

We have included the statement. Thank you for pointing this out.

*e) Pag.10 line 7 from bottom: survival in circulation for 7 days (estimate t1/2 about 2.5 days) in macrophage-depleted NOD/SCID mice seems to be very low. What is the t1/2 of unprocessed (native) human RBC under the same conditions? Please provide comparison data to discriminate the role of the host vs the role of the *in vitro* differentiation of transduced erythroblasts.*

We are sorry for the confusion. It was re-written to “we can detect the cells in the circulation for at least 7 days”. After 7 days, the signal is below our detection limit. The control experiment suggested here was done in a recent paper, *Haematologica* 102, 476-483 (2017), and we have now referenced this paper in our manuscript

f) Pag.11 line 1 from top: please comment on the reduced expression on human RBC vs murine RBC of the constructs.

We have commented on this in the last paragraph in the Results section.

g) DISCUSSION: the discussion section is very speculative and several conclusions are not fully supported by data. Two key issues should be mentioned in the discussion at least, the safety issue including possible alloimmunization (I have already commented above) and the transfer in the recipient of retroviral transduced cells. While the Authors have documented the expansion and

maturation in vitro to be quite efficient, still a significant number of nucleated cells are present (about 30% of engineered human RBC, pag.11 line three from top).
We admit that there is only 65~75% enucleation of engineered RBCs, and we have now included the safety issues and possible solutions in the second to last paragraph of the discussion.

h) Pag.13 line 5-6: biological production of recombinant protein is certainly not an issue nowadays!

This statement should have been removed prior to our submission. We are sorry for making this mistake.

i) Pag.13 line 7 from bottom: clearance of BoNT/A bound to RBC is a relevant issue and should be evaluate in more details since the antibody is not neutralizing the toxin and the accumulation of the same in some compartments could represent an important issue.

This is indeed an important issue and yet, as noted above, we do not have a good explanation for the mechanism of clearance of aged normal red cells, let alone the clearance of BoNT/A bound to RBCs. As mentioned earlier, we could not test this easily since the flow cytometry detection capacity is lower than the necessary resolution needed to identify the final resting place of these engineered red cells *in vivo*. Therefore, we are developing a more sensitive biosynthetic labeling of our engineered RBCs to trace the phenomenon *in vivo*. In spite of that, our data suggests that the mice that were protected by GPA-VNA/A RBCs are alive and healthy for months after challenges, implying that accumulation of toxin in some cell or subcellular compartment is a negligible factor.

j) Pag.14 speculations about the use of CR1 as a coupling site for VHH, the use of more than three VHH domains on the same cell or the expression of functional cargos inside the RBC are speculations that should be contained in few lines or less.

We deleted the speculation concerning CR1 and sentences mentioning the trimer.

k) Pag.15, last line from bottom: the estimate of 26 days survival of cRBCs should be taken carefully since it represent only one donor, the labelled cells were retics and the Cr51 elution was not estimated but extrapolate from other studies. As a consequence, the last sentence in the Discussion should mention the need of additional data to confirm the expectations!

We agree with your concerns and we now address them in the last paragraph of the discussion.

REVIEWERS' COMMENTS:

Reviewer #1 (Remarks to the Author):

The authors tried to address the points I raised. I have no further comments

Reviewer #3 (Remarks to the Author):

The points raised in the previous round of review have been satisfactorily addressed.

Dear reviewers,

Thank you for considering our manuscript for publication in Nature Communications and thank you for reviewing our revised manuscript. We appreciate the points you have raised and the suggestions you have made.

REVIEWERS' COMMENTS:

Reviewer #1 (Remarks to the Author):

The authors tried to address the points I raised. I have no further comments

Reviewer #3 (Remarks to the Author):

The points raised in the previous round of review have been satisfactorily addressed.